# CoIRL-AD: Collaborative-Competitive Imitation-Reinforcement Learning in Latent World Models for Autonomous Driving

**Xiaoji Zheng** [* 1]  **Ziyuan Yang** [* † 2]  **Yanhao Chen** [† 3]  **Yuhang Peng** [† 4]  **Yuanrong Tang** [1]  **Gengyuan Liu** [1]
**Bokui Chen** [1]  **Jiangtao Gong** [5]

## Abstract

End-to-end autonomous driving models trained with imitation learning (IL) often generalize poorly, particularly in long-tail scenarios where expert demonstrations are sparse. Reinforcement learning (RL) can provide complementary task-level supervision, but applying RL to real-world autonomous driving is challenging in offline settings without interactive simulators, where datasets are dominated by expert actions and provide limited behavioral diversity. We propose CoIRL-AD, a competitive dual-policy framework that integrates IL and RL under a unified offline training regime. CoIRL-AD decouples imitation and reward optimization into separate actors to alleviate objective conflicts, uses imagined future rollouts for long-horizon reward estimation, and introduces a competition mechanism that selectively transfers beneficial behaviors while keeping RL anchored to expert-like driving. Experiments on the nuScenes benchmark show that CoIRL-AD consistently improves robustness over strong IL-based baselines, with especially large gains in cross-city generalization and long-tail scenarios. Code is available at: `https://github.com/SEU-zxj/CoIRL-AD`.

## 1. Introduction

End-to-end (E2E) learning has become the mainstream paradigm in autonomous driving (Hu et al., 2023; Jiang et al., 2023; Weng et al., 2024). Unlike modular pipelines,

E2E models allow gradients to propagate across perception, prediction, and planning, enabling all components to be optimized toward the final driving objective.

Most existing E2E approaches rely on imitation learning (IL), where models are trained to mimic expert demonstrations. In practice, IL is often formulated as supervised learning (SL), with model outputs directly supervised by expert trajectories. However, supervised imitation is optimized on a fixed data distribution, while embodied driving agents induce their own state distributions at deployment. Small prediction errors can shift the agent into unseen states and accumulate over time, causing IL-based agents to generalize poorly and struggle in long-tail scenarios.

Reinforcement learning (RL) provides a natural way to address these limitations, as it can learn from reward signals even in scenarios where human demonstrations are unavailable. This motivates integrating RL into IL-based E2E driving systems to improve generalization and long-tail performance. Since E2E driving models are commonly trained and evaluated on offline real-world datasets (Caesar et al., 2020; Dauner et al., 2024), we restrict RL to the offline setting as well. Unlike prior RL-based driving methods that rely on simulators (Dosovitskiy et al., 2017), we do not use an external simulator, as introducing one would convert the problem from offline RL to online RL. Under this setting, we study the following research question: **In offline real-world autonomous driving without an external simulator, how can reinforcement learning be used to improve performance?**

We first design an RL framework tailored to offline E2E driving. Unlike traditional offline RL benchmarks that contain diverse suboptimal trajectories, real-world driving datasets are dominated by near-optimal expert demonstrations. Such expert-dominated data provide limited coverage of non-expert behaviors, making it difficult to learn value differences among alternative actions. To address this issue, we introduce a group sampling strategy inspired by GRPO (Shao et al., 2024), which increases behavioral diversity by sampling multiple action candidates. For each sampled trajectory, we compute rule-based rewards using human-annotated maps and bounding boxes available in the dataset.

---

[*]Equal contribution [†]Work done during internships at Institute for AI Industry Research (AIR), Tsinghua University [1]Tsinghua University [2]University of Washington [3]Beijing Jiaotong University [4]The Hong Kong Polytechnic University [5]Shanghai Jiao Tong University. Correspondence to: Bokui Chen <chenbk@tsinghua.edu.cn>, Jiangtao Gong <gongjiangtao@sjtu.edu.cn>.

*Proceedings of the 43rd International Conference on Machine Learning*, Seoul, South Korea. PMLR 306, 2026. Copyright 2026 by the author(s).

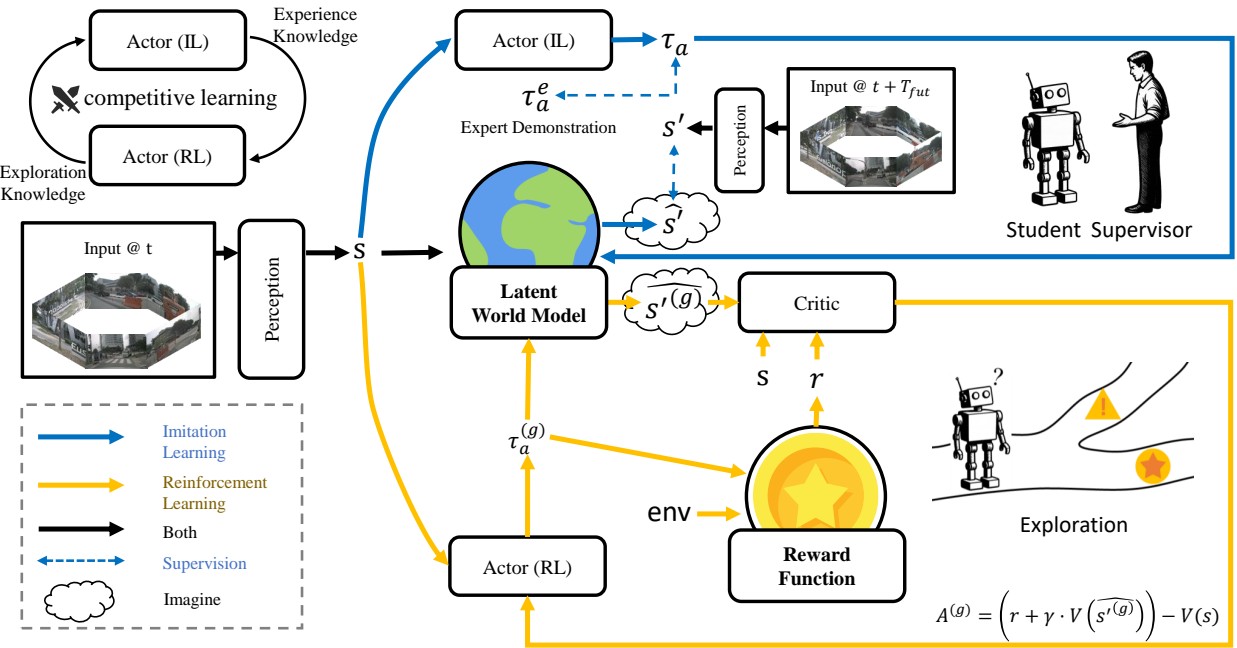

*Figure 1.* **Overview of CoIRL-AD.** CoIRL-AD adopts a dual-policy architecture that integrates imitation learning (IL) and reinforcement learning (RL) through a shared latent world model. In each iteration, the IL actor and RL actor are trained in parallel. The latent world model is learned during the IL phase and then used in the RL phase, where only the RL actor and critic are updated. For exploration, the RL actor samples multiple action sequences, predicts future states via the latent world model, and evaluates them with rule-based reward functions. The critic assigns advantages to each sequence based on the imagined trajectories and rewards. To promote interaction, a competitive learning mechanism exchanges knowledge between the IL and RL actors.

To optimize long-horizon cumulative rewards rather than immediate rewards, we introduce a latent world model that provides imagined future rollouts conditioned on sampled trajectories.

We then investigate how to effectively integrate IL and RL under this offline setting. Since the latent world model is trained on expert data, its predictions can be biased for out-of-distribution actions. During RL optimization, such biases may lead to over-optimistic value estimates and reinforce suboptimal behaviors, resulting in unstable training and performance degradation. This makes RL-only learning and conventional two-stage approaches that fine-tune IL policies with RL vulnerable to instability in our setting. While prior one-stage methods jointly optimize IL and RL objectives, they suffer from gradient conflicts due to the mismatch between behavior cloning and reward maximization objectives (Gao et al., 2025). To address the gradient conflict issue, we decouple IL and RL into two separate actors, isolating imitation learning from RL exploration noise. On top of this, to enable interaction between the two actors, we propose a competitive learning mechanism that selectively transfers beneficial behaviors while anchoring RL learning to IL performance, allowing the RL policy to recover from degraded updates.

Finally, experimental results show that RL provides larger

benefits in challenging scenarios with limited or missing demonstrations, while its impact is smaller in well-covered scenarios. Our method achieves significant improvements in cross-city generalization and long-tail scenarios, demonstrating the effectiveness of competitive IL/RL integration for offline E2E autonomous driving.

The remainder of the paper is organized as follows. We first describe actor modeling and the stochastic policy parameterization used for reinforcement learning (Section 3.1), followed by the backward planning design that improves RL optimization (Section 3.2). We then present the reinforcement learning objective in detail (Section 3.3) and introduce the competitive dual-policy mechanism that selectively transfers beneficial behaviors between the IL and RL actors (Section 3.4).

## 2. Related Work

### 2.1. End-to-end Autonomous Driving

End-to-end autonomous driving methods replace traditional modular pipelines with unified models optimized directly for driving objectives. UniAD (Hu et al., 2023) demonstrates the potential of this paradigm by integrating perception and planning within a single framework. VAD (Jiang et al., 2023) vectorizes scene representations to improve

inference efficiency, while PARA-Drive (Weng et al., 2024) decomposes the traditional pipeline and searches for effective end-to-end architectures. Recent work also incorporates world models into driving systems. LAW (Li et al., 2025b) and World4Drive (Zheng et al., 2025) predict future visual latents to improve temporal understanding. SSR (Li & Cui, 2025) uses sparse tokens to represent dense BEV features and similarly predicts future features for scene comprehension. WoTE (Li et al., 2025c) leverages a world model to predict future states, enabling online trajectory evaluation and selection. In contrast, our approach uses a latent world model as a reactive simulator for policy learning: the RL actor interacts with the world model to imagine future scene transitions and optimize reward-driven behavior.

## 2.2. RL in Autonomous Driving

Reinforcement learning has been widely studied in autonomous driving. Roach (Zhang et al., 2021) trains an RL expert that maps BEV inputs to driving actions and subsequently uses it as a teacher for a student policy. VLM-RL (Huang et al., 2024) leverages a vision-language model (VLM) to generate reward signals for RL. Think2Drive (Li et al., 2024) integrates DreamerV3 (Hafner et al., 2023) to train an expert agent and achieves strong performance in CARLA. AdaWM (Wang et al., 2025) analyzes performance degradation in driving agents and proposes to selectively update the actor or world model. Imagine2Drive (Garg & Krishna, 2024) combines a video world model (Gao et al., 2024) with a diffusion-based policy, achieving impressive performance in CARLA. Urban Driver (Scheel et al., 2022) constructs a differentiable simulator from perception outputs and high-definition maps to support efficient policy learning. Inspired by these approaches, we also perform RL with a learned world model. However, instead of relying on an external simulator such as CARLA, we train the world model directly from offline real-world driving data, avoiding simulator-specific assumptions and reducing the sim-to-real gap. Moreover, we integrate model-based RL with imitation learning to improve training stability and data efficiency in offline E2E autonomous driving.

## 2.3. Combining Imitation Learning and Reinforcement Learning

Integrating imitation learning and reinforcement learning has attracted growing attention in autonomous driving and related domains. Existing approaches can be broadly categorized into two paradigms. The first paradigm adopts a two-stage strategy, where models are first pretrained with IL and then refined using RL. For example, AutoVLA (Zhou et al., 2026) performs supervised fine-tuning before applying GRPO (Shao et al., 2024) for further improvement. RAD (Gao et al., 2026) constructs a large-scale 3D environment and leverages RL fine-tuning after IL pretraining, while Tra-

jHF (Li et al., 2025a) combines IL fine-tuning with RLHF using large-scale preference data. Similar two-stage or reward-based training strategies have also been explored in recent LLM and LMM studies, where supervised fine-tuning and RL exhibit different generalization behaviors (Wu et al., 2026; Zhang et al., 2026; Peng et al., 2025). The second paradigm follows a one-stage strategy that jointly optimizes IL and RL objectives. ReCogDrive (Li et al., 2026) incorporates imitation and RL losses in a simulator to encourage safer trajectory exploration, while BC-SAC (Lu et al., 2023) optimizes a weighted combination of behavior cloning and RL objectives. However, optimizing imitation and reward maximization within a single policy can introduce objective conflicts (Gao et al., 2025). Our approach also performs IL and RL within a unified training process, but differs from prior one-stage methods by introducing a dual-policy competitive framework. By assigning IL and RL to separate actors, CoIRL-AD alleviates objective conflicts while allowing the two policies to exchange beneficial behaviors through competition.

## 3. Method

### 3.1. Actor Modeling

Given current observation $o$ (usually images captured by cameras), the perception module encodes it into latent state $s \in \mathbb{R}^{B \times N_t \times D}$, where $B$ is the batch size, $N_t$ is the number of tokens for each latent state, and $D$ is the feature dimension. For planning, a waypoint query $Q_w \in \mathbb{R}^{B \times n \times D}$ is employed to extract the waypoint features $s_w = \{s_{w,1}, s_{w,2}, ..., s_{w,n}\} \in \mathbb{R}^{B \times n \times D}$ through cross-attention, where $n$ denotes the number of waypoints in a trajectory. The planning head then decodes the waypoint features $s_w$ into an action sequence $\tau_a = \{a_1, a_2, ..., a_n\}$, where $a_i \in \mathbb{R}^2$ and represents the value of x and y of the agent's action. Using the provided expert action demonstrations $\tau_a^e$ as labels, imitation learning applies an L1 loss $L_{imi}$ to supervise output.

$$s_w = \text{CrossAttn}(q = Q_w, k = s, v = s), \quad (1)$$

$$\tau_a = \text{PlanningHead}(s_w), \quad (2)$$

$$L_{imi} = ||\tau_a - \tau_a^e||. \quad (3)$$

To endow the model with predictive capability, we use a world model to predict future states. Unlike pixel-level generative world models, our model operates in latent space to reduce task complexity. Specifically, given the current state $s$ and action sequence $\tau_a$, the world model predicts the future state $\hat{s}'$:

$$\hat{s}' = \text{LatentWorldModel}(s, \tau_a). \quad (4)$$

Meanwhile, the perception module encodes the next observation $o'$ into the ground-truth state $s'$. The latent world

model is trained in a self-supervised manner using mean squared error (MSE)[1]. The overall imitation learning loss $L_{IL}$ combines $L_{wm}$ and $L_{imi}$, where $\alpha$ is a hyperparameter set to 0.2 following LAW (Li et al., 2025b):

$$L_{wm} = \text{MSELoss}(s', \hat{s}'), \tag{5}$$

$$L_{IL} = L_{imi} + \alpha \cdot L_{wm}. \tag{6}$$

## 3.2. Backward Planning

In practice, the planning head predicts $\tau_a$ in a single forward pass. This design overlooks dependencies among the steps in $\tau_a$. A natural extension is to adopt a self-attention layer with a causal mask to introduce temporal causality. The policy for $a_i$ is formulated as $\pi_i(a_i|s_{w,1}, ..., s_{w,i}) = \pi(a_i|s_{w,j\leq i})$.

While this forward-causal design appears intuitive, human driving behavior suggests an alternative perspective. Drivers typically decide **where to go** before committing to low-level actions. Moreover, in real-world deployment, only the first action is executed before replanning, making earlier actions matter more.

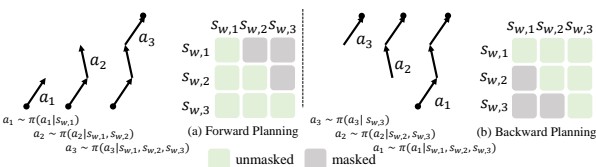

*Figure 2.* Comparison between forward planning with a causal mask and backward planning with an inverse causal mask.

Motivated by these insights, as shown in Fig. 2, we explore a counterintuitive alternative **backward planning** (inverse causality), where the i-th action is conditioned on the current and future waypoint features:

$$\pi_i(a_i|s_{w,i}, ..., s_{w,n}) = \pi_i(a_i|s_{w,j\geq i}) \tag{7}$$

This formulation provides early actions with richer contextual information while leaving later actions less constrained. Inverse causality changes only the conditioning order and does not directly affect the smoothness of the final trajectory. Prior evidence from embodied AI (Liu et al., 2025) further supports this goal-to-action reasoning paradigm, and our experiments in Tab. 3 show that this goal-oriented design benefits the RL actor, whose objective is to maximize cumulative rewards.

---

[1]In implementation, we follow LAW's setting, where the next observation corresponds to the observation 1.5 seconds in the future.

## 3.3. Reinforcement Learning

RL requires rewards to evaluate the quality of explored trajectories. Given a predicted action sequence $\tau_a$, the corresponding position sequence is obtained via cumulative summation: $\tau_{pos} = \{a_1, a_1 + a_2, ..., \sum_i a_i\}$. We use two components to define the reward: an imitation reward $r_{imi}$ and a collision reward $r_{col}$:

$$r_{imi}^{(i)} = e^{-||a_i - a_i^e||_2},$$
$$r_{col}^{(i)} = 1 - \text{CollisionDetection}(\tau_{pos}^{(i)}, \text{env}). \tag{8}$$

Here, env denotes static map information and the trajectories of dynamic agents, implemented as non-reactive simulation for simplicity. The final reward for step $i$ is defined as:

$$r_i = r_{col}^{(i)} \cdot r_{imi}^{(i)}. \tag{9}$$

On other offline datasets such as Navsim (Dauner et al., 2024), richer reward signals can be constructed to evaluate trajectory quality, including comfort-related metrics and time-to-collision, among others.

The deterministic action sequence produced by the model cannot be directly optimized with standard RL objectives, which typically require probabilistic policies. To address this, we model uncertainty in the action sequence. In addition to the planning head, which outputs the mean value $\mu_i$ for each action, we use a stochastic head to output the uncertainty of each action:

$$\tau_\sigma = \{\sigma_1, \sigma_2, ..., \sigma_n\} = \text{StochasticHead}(s_w), \tag{10}$$

where $\sigma_i$ is the standard deviation for $a_i$. We model each action with a Gaussian distribution and use a diagonal covariance matrix. The policy for action $a_i$ is then formulated as:

$$\pi_i(a_i|s_{w,j\geq i}) = \mathcal{N}(\mu_i, \sigma_i^2 I), \tag{11}$$

where $I$ denotes the identity matrix. Given the offline imitation dataset $\{(s, \tau_a^e, \tau_r^e, s'), ...\}$, a Gaussian log-likelihood loss can be used to clone expert behavior: $L_{bc} = -\sum_{i=1}^n \log \pi_i(a_i^e|s_{w,j\geq i})$. However, we want the RL actor to explore and learn from both successful and unsuccessful candidate behaviors. Inspired by GRPO (Shao et al., 2024), we introduce exploration by sampling $G$ trajectories from the policy and computing their corresponding reward sequences with the rule-based reward function. Each action sequence in the group, together with its reward sequence, is formulated as:

$$\tau_a^{(g)} = \{a_1^{(g)}, a_2^{(g)}, ..., a_n^{(g)}\},$$
$$\tau_r^{(g)} = \{r_1^{(g)}, r_2^{(g)}, ..., r_n^{(g)}\}, \tag{12}$$

where $a_i^{(g)} \sim \pi_i(a_i^{(g)}|s_{w,j\geq i})$ is the $i$-th action of $g$-th trajectory in the group. For each trajectory, we compute its

total reward and normalize it within the group using Z-score normalization to obtain the advantage. The naive policy gradient loss with group sampling (naive PGGS) is computed by:

$$L_{actor} = -\frac{1}{G}\sum_{g=1}^{G} A^{(g)} \left(\sum_{i=1}^{n} \log \pi_i(a_i^{(g)}|s_{w,j\geq i})\right),$$

$$A^{(g)} = \frac{\Sigma\tau_r^{(g)} - \text{mean}(\Sigma\tau_r^{(1)},...,\Sigma\tau_r^{(G)})}{\text{std}(\Sigma\tau_r^{(1)},...,\Sigma\tau_r^{(G)})}. \quad (13)$$

To extend the advantage estimation to long-term rewards, we train a critic model $V$ to output the value of both current state $s$ and the next state $s'$. Since the offline dataset does not provide next states for sampled trajectories, we leverage the latent world model to generate future states:

$$s'^{(g)} = \text{LatentWorldModel}(s, \tau_a^{(g)}). \quad (14)$$

The long-term advantage $A_{long}^{(g)}$ is computed by:

$$A_{long}^{(g)} = \left(\sum r^{(g)} + \gamma V(s'^{\hat{(g)}})\right) - V(s), \quad (15)$$

in which $\gamma$ is a discount factor. We further apply Z-score normalization to $A_{long}^{(g)}$ within each group and denote the normalized advantage as the critic advantage $A_{cri}^{(g)}$. During training, the actor and critic are then jointly optimized, and this is the method of actor + dreaming critic with group sampling (ADCGS):

$$L_{act} = -\frac{1}{G}\sum_{g=1}^{G} A_{cri}^{(g)} \left(\sum_{i=1}^{n} \log \pi_i(a_i^{(g)}|s_{w,j\geq i})\right),$$

$$L_{cri} = \frac{1}{G}\sum_{g=1}^{G} \left[V(s) - \left(\sum \tau_r^{(g)} + \gamma V(s'^{\hat{(g)}})\right)\right]^2. \quad (16)$$

Since pure RL with our reward is difficult to optimize reliably (see Tab. 4), we incorporate a small behavior cloning term $L_{bc}$ with coefficient $\beta$ as a regularizer, inspired by the role of KL regularization in policy optimization. This auxiliary loss provides weak expert guidance during the competitive phase, where the RL actor would otherwise operate without direct expert supervision. In our experiments, omitting this term still yields reasonable performance, but adding it with a small $\beta$ further improves the results (see Tab. 6).

$$L_{RL} = L_{act} + L_{cri} + \beta \cdot L_{bc}. \quad (17)$$

In practice, since actions in a sampled sequence are drawn independently from different Gaussian policies, the resulting position trajectory $\tau_{pos}$ may lack smoothness. To address this, we adopt a step-aware mechanism: within each sampled sequence, only one action is stochastic, while the remaining actions are set to the mode of their respective policies, ensuring a smoother $\tau_{pos}$. The detailed algorithm and visualizations are provided in Appendix A. To further stabilize critic learning, we employ the two-critic trick, where a reference critic maintains an exponential moving average (EMA) of the learning critic.

### 3.4. Dual-policy Learning Framework

During training, to avoid gradient conflicts from jointly optimizing IL and RL losses in a single policy, we decouple the planning module into an IL actor and an RL actor, optimized by $L_{IL}$ and $L_{RL}$, respectively. To encourage interaction between the two actors and selectively transfer beneficial behaviors, we allow the actors to compete and share information.

To balance the contributions of the imitation learning (IL) actor and the reinforcement learning (RL) actor, we periodically compare their performance every $k$ iterations. The comparison is based on the difference between the accumulative reward scores achieved by the IL actor and the RL actor ($\Delta r_{acc}$), together with two thresholds ($\lambda_{min}$, $\lambda_{max}$) that measure the size of the score gap and a hyperparameter $p$ that controls the degree of parameter interpolation.

Based on the score difference $\Delta r_{acc}$, we adaptively update the loser actor's parameters: 1) If the two actors perform similarly, we keep both unchanged; 2) If the performance gap is moderate, we apply soft merging to gradually transfer knowledge from the winner to the loser (i.e. *loser.weight* := *loser.weight* $\cdot p$ + *winner.weight* $\cdot (1-p)$); 3) If the gap is large, we directly replace the loser's parameters with the winner's.

This adaptive mechanism enables stable cooperation between the IL and RL actors, preventing premature dominance while allowing faster convergence once one actor becomes consistently superior. Hyperparameter details are provided in Appendix D. The ablation results in Tab. 4 and the analysis in Fig. 5 further show that competition between the IL and RL actors leads to better results.

## 4. Experiments

### 4.1. Benchmarks

We conduct experiments on the offline autonomous driving dataset nuScenes (Caesar et al., 2020). Detailed benchmark descriptions and implementation details are provided in Appendix B and Appendix C.

### 4.2. Main Results

The results on nuScenes are summarized in Tab. 1. We follow the evaluation protocol of (Jiang et al., 2023), reporting the average L2 displacement error and collision rate over 1s, 2s, and 3s prediction horizons. We evaluate our method on

the LAW (Li et al., 2025b) framework.

On nuScenes, CoIRL-AD consistently outperforms the baseline LAW across both L2 error and collision rate. Notably, even without temporal augmentation (which is commonly adopted by prior methods, provide not only current observation, but also 2-3 past observation to the model), CoIRL-AD already achieves the lowest collision rate. After enabling temporal augmentation, the L2 error is further reduced substantially. Overall, our method achieves the best performance on the combined L2 · Col metric, demonstrating its effectiveness in both accuracy and safety. Moreover, CoIRL-AD shares the same inference architecture as the baseline and introduces no additional inference latency. Detailed training cost and inference latency analyses are provided in Appendix E.1 and E.2.

Despite these improvements, we note that RL is expected to provide greater benefits in scenarios where expert demonstrations are sparse or insufficient. Since nuScenes contains a large proportion of well-covered, relatively easy driving scenarios, improvements on challenging cases can be diluted in average metrics. To validate this intuition, we further evaluate our method on more challenging settings, including cross-city generalization and long-tail scenarios.

**Generalization Ability.** We evaluate cross-city generalization by training on nuScenes-Singapore and testing on nuScenes-Boston (Tab. 2 and Fig. 12). The baseline LAW exhibits significant performance degradation when evaluated in an unseen city. In contrast, CoIRL-AD demonstrates markedly improved robustness, which we attribute to the exploration capability introduced by the RL actor.

**Performance under Long-Tail Scenarios.** To assess performance on rare and challenging cases, we construct two evaluation subsets based on the baseline's performance: one consisting of scenarios with high L2 error, and another with high collision rate. Results in Fig. 3 show that incorporating reinforcement learning leads to substantial improvements over the baseline in both subsets. Details on subset construction and complete quantitative results are provided in Appendix E.3.

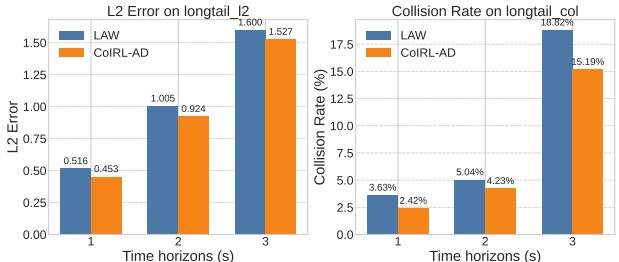

*Figure 3.* Performance under long-tail scenarios

Overall, these results indicate that reinforcement learning is particularly beneficial in challenging and under-represented

driving scenarios, and that stable integration with imitation learning is crucial for realizing these gains in offline, real-world settings.

### 4.3. Ablation Study

**Effect of Causality.** To examine the role of inverse causality, we conduct three ablation experiments on CoIRL-AD without temporal augmentation. We vary the auto-regression (AR) attention mask used for $s_w$ in the self-attention layer as follows: (1) no causal mask; (2) standard causal mask (AR)[2]; and (3) inverse causal mask (inv. AR)[3]. Results are reported in Tab. 3.

The results suggest three observations. First, inverse causal masking primarily benefits RL exploration: applying *inv. AR* to the pure IL baseline degrades performance, while using it within CoIRL-AD improves both L2 error and collision rate. Second, aligned representations are important for competitive learning. When the IL and RL actors use mismatched masks, the shared modules learn inconsistent representations, weakening knowledge transfer and reducing performance toward the baseline. Third, RL exploration must be stabilized. Removing the competitive mechanism causes the RL actor to explore aggressively and suffer from value overestimation, leading to degraded performance.

The best L2 · Col score in Row 6 indicates a strong synergy between inverse causal masking and competitive learning. The inverse causal mask helps the RL actor discover better trajectories, while the competitive mechanism anchors this exploration through aligned IL/RL representations.

**Effect of the Latent World Model in RL.** As shown in Tab. 1, CoIRL-AD consistently outperforms its variant without a latent world model (w/o wm), with particularly significant reductions in collision rate at the 3-second horizon. This demonstrates that incorporating a learned world model as a reactive simulator for reinforcement learning is both effective and critical.

**Integration of IL and RL.** We compare multiple strategies for integrating imitation learning and reinforcement learning: (i) *loss merging*, jointly optimizing $L_{IL} + L_{RL}$; (ii) *IL–RL interval*, alternating optimization between $L_{IL}$ and $L_{RL}$; (iii) *two-stage*, pre-training with $L_{IL}$ followed by fine-tuning with $L_{RL}$; and (iv) *decoupled actors*, where IL and RL actors are optimized separately, optionally with competitive interaction ("comp"). Temporal augmentation is not used in these experiments. Results are shown in Tab. 4.

Among all variants, only the *decoupled actors with competition* strategy improves both L2 error and collision rate

---

[2]Causal mask: *torch.triu(torch.ones(n,n), diagonal=1).bool()*
[3]Inverse causal mask: *torch.tril(torch.ones(n,n), diagonal=-1).bool()*

*Table 1.* **Comparison of state-of-the-art methods on the nuScenes dataset.** Perception-based Methods are those models who add detection, mapping, tacking heads, and use labeled data to supervise the output results while perception-free methods are not. [†]Methods that use temporal augmentation. 'w/o wm' means we do not use the latent world model in RL process. The best results are shown in **bold**, and the second-best results are underlined.

| Method | L2 (m) ↓ | | | | Col (%) ↓ | | | | L2 · Col ↓ |
| --- | --- | --- | --- | --- | --- | --- | --- | --- | --- |
| | 1s | 2s | 3s | Avg. | 1s | 2s | 3s | Avg. | Avg. |
| **Perception-based Methods** | | | | | | | | | |
| ST-P3[†] | 1.33 | 2.11 | 2.90 | 2.11 | 0.23 | 0.62 | 1.27 | 0.71 | 1.50 |
| UniAD[†] | 0.48 | 0.96 | 1.65 | 1.03 | 0.05 | 0.17 | 0.71 | 0.31 | 0.32 |
| VAD[†] | 0.41 | 0.70 | 1.05 | 0.72 | 0.07 | 0.17 | 0.41 | 0.22 | 0.16 |
| PARA-Drive[†] | 0.25 | 0.46 | 0.74 | 0.48 | 0.14 | 0.23 | 0.39 | 0.25 | 0.12 |
| GenAD[†] | 0.28 | 0.49 | 0.78 | 0.52 | 0.08 | 0.14 | 0.34 | 0.19 | 0.10 |
| LAW[†] | 0.24 | 0.46 | 0.76 | 0.49 | 0.08 | 0.10 | 0.39 | 0.19 | 0.09 |
| **Perception-free Methods** | | | | | | | | | |
| BEV-Planner[†] | 0.30 | 0.52 | 0.83 | 0.55 | 0.10 | 0.37 | 1.30 | 0.59 | 0.32 |
| SSR[†] | 0.18 | 0.35 | 0.62 | **0.38** | 0.48 | 0.45 | 0.51 | 0.48 | 0.18 |
| LAW | 0.32 | 0.62 | 1.03 | 0.66 | 0.08 | 0.13 | 0.46 | 0.22 | 0.15 |
| CoIRL-AD (w/o wm) | 0.31 | 0.61 | 1.01 | 0.65 | 0 | 0.10 | 0.51 | 0.20 | 0.13 |
| CoIRL-AD | 0.29 | 0.59 | 1.00 | 0.63 | 0.06 | 0.10 | 0.37 | 0.18 | 0.11 |
| CoIRL-AD[†] | 0.23 | 0.42 | 0.70 | 0.45 | 0.10 | 0.12 | 0.30 | **0.17** | **0.08** |

*Table 2.* **Cross-city generalization ability on nuScenes dataset.**

| Method | L2 (m) ↓ | | | | Col (%) ↓ | | | | L2 · Col ↓ |
| --- | --- | --- | --- | --- | --- | --- | --- | --- | --- |
| | 1s | 2s | 3s | Avg. | 1s | 2s | 3s | Avg. | Avg. |
| LAW | 0.45 | 0.89 | 1.46 | 0.93 | 0.13 | 0.43 | 1.50 | 0.69 | 0.64 |
| CoIRL-AD | 0.33 | 0.65 | 1.13 | **0.70** | 0.04 | 0.15 | 0.46 | **0.22** | **0.15** (↓ 77%) |

*Table 3.* **Effect of causality on performance**

| Method | L2 (m) ↓ | | | | Collision Rate (%) ↓ | | | | L2 · Col ↓ |
| --- | --- | --- | --- | --- | --- | --- | --- | --- | --- |
| | 1s | 2s | 3s | Avg. | 1s | 2s | 3s | Avg. | Avg. |
| LAW | 0.32 | 0.63 | 1.03 | 0.66 | 0.09 | 0.12 | 0.46 | 0.22 | 0.15 |
| LAW (inv. AR) | 0.37 | 0.70 | 1.13 | 0.73 | 0.11 | 0.13 | 0.50 | 0.25 | 0.18 |
| CoIRL-AD (no mask) | 0.30 | 0.60 | 1.01 | 0.64 | 0.06 | 0.14 | 0.53 | 0.24 | 0.15 |
| CoIRL-AD (AR) | 0.36 | 0.69 | 1.12 | 0.72 | 0.04 | 0.15 | 0.55 | 0.25 | 0.18 |
| CoIRL-AD (IL AR + RL inv. AR) | 0.32 | 0.62 | 1.04 | 0.66 | 0.08 | 0.12 | 0.48 | 0.23 | 0.15 |
| CoIRL-AD (inv. AR) | 0.29 | 0.59 | 1.00 | **0.63** | 0.06 | 0.10 | 0.37 | **0.18** | **0.11** |
| CoIRL-AD (inv. AR, w/o comp) | 0.36 | 0.68 | 1.11 | 0.72 | 0.11 | 0.17 | 0.60 | 0.29 | 0.21 |

*Table 4.* **Comparison of different IL–RL integration strategies.**

| Description | L2 (m) ↓ | | | | Collision Rate (%) ↓ | | | | L2 · Col ↓ |
| --- | --- | --- | --- | --- | --- | --- | --- | --- | --- |
| | 1s | 2s | 3s | Avg. | 1s | 2s | 3s | Avg. | Avg. |
| LAW (pure IL) | 0.32 | 0.63 | 1.03 | 0.66 | 0.09 | 0.12 | 0.46 | 0.22 | 0.15 |
| pure RL | 3.92 | 6.55 | 9.18 | 6.55 | 2.75 | 4.87 | 7.72 | 4.93 | 32.29 |
| loss merging | 0.38 | 0.73 | 1.17 | 0.76 | 0.03 | 0.12 | 0.54 | 0.23 | 0.17 |
| IL-RL interval | 0.31 | 0.63 | 1.07 | 0.68 | 0.12 | 0.17 | 0.54 | 0.28 | 0.19 |
| two-stage | 2.43 | 4.21 | 6.03 | 4.22 | 2.29 | 4.13 | 6.53 | 4.32 | 18.23 |
| decouple, w/o comp | 0.36 | 0.68 | 1.11 | 0.72 | 0.11 | 0.17 | 0.60 | 0.29 | 0.21 |
| decouple, w/ comp | 0.29 | 0.59 | 1.00 | **0.63** | 0.06 | 0.10 | 0.37 | **0.18** | **0.11** |

over the baseline. This result is noteworthy, as two-stage IL–RL transfer has proven effective in other domains (e.g., DeepSeek-R1 (Guo et al., 2025)). We attribute the limited effectiveness of other strategies to the offline training setting. Since the latent world model is trained on expert data, its predictions can be biased for out-of-distribution actions. During RL optimization, such bias may lead to over-optimistic value estimates, reinforcing suboptimal behaviors and causing unstable training, even when initialized from expert demonstrations. In contrast, the competitive decoupled design mitigates these issues by using the IL actor as a stabilizing anchor. When the RL actor degrades due to over-optimistic value estimates or exploitation of world-model errors, the competition mechanism transfers information from the stronger actor, improving stability and yielding measurable performance gains.

## 5. Analysis

### 5.1. Competition Analysis

The last two rows of Tab. 4 show that the competitive learning mechanism helps the IL and RL actors interact and ultimately learn a stronger model. We next analyze how this mechanism affects training.

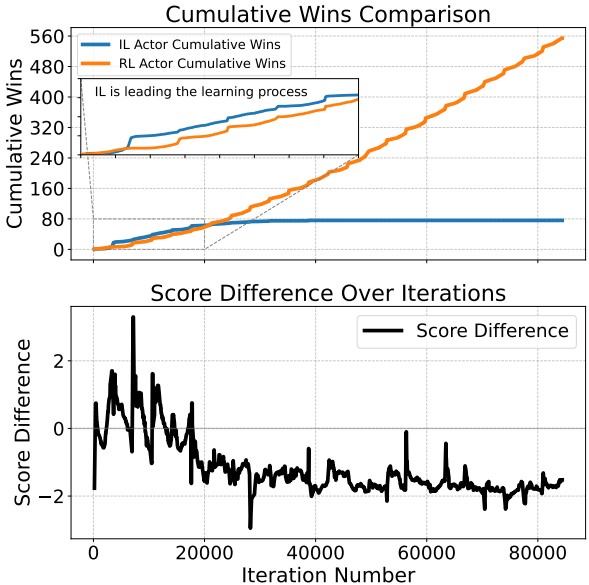

*Figure 4.* Accumulated wins (top) and score difference (bottom) across training iterations.

By tracking accumulated wins and score differences (IL score – RL score) over iterations, we observe the following pattern (see Fig. 4). In the early stage (<20k iterations), the IL actor achieves more wins and higher scores, indicating that IL initially leads the learning process. At this point, the latent world model provides limited useful signal, leaving the RL actor unable to extract reliable guidance from explo-

ration. Afterward, as the latent world model starts to encode the underlying driving dynamics, the RL actor progressively learns fundamental driving behaviors through exploration. Its group-sampling-based exploration then becomes more effective than simply imitating expert trajectories. Consequently, the RL actor achieves higher scores and dominates in later training. This progression resembles the two-stage paradigm, but with a key difference: IL and RL are trained jointly. Even though IL loses more frequently in later stages, its gradients continue to benefit shared components such as the perception module.

To further analyze why the competitive mechanism improves learning, we log the evolution of the critic's value estimates (see Fig. 5) and the policy divergence between the IL and RL actors (see Fig. 6), measured by the L2 distance between their modal trajectories.

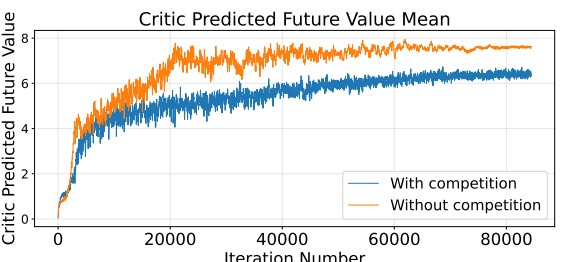

*Figure 5.* Mean critic value estimates for future states predicted by the world model.

Without the competitive mechanism, the value estimate for the imagined future state $V(\hat{s_{t+1}})$ exhibits severe overestimation and grows rapidly. With the mechanism active, the value estimate plateaus. This suggests that unconstrained offline RL can exploit world-model inaccuracies to produce artificially high expected returns, while competition helps suppress this behavior.

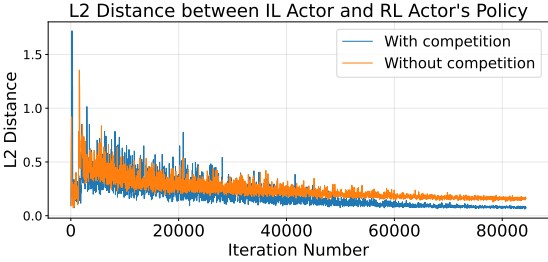

*Figure 6.* L2 distance between the modal trajectories of the IL actor and RL actor.

Driven by the overestimated values above, the unconstrained RL actor tends to select dangerous or highly unconventional actions to pursue hallucinated rewards. This shifts its policy away from the expert distribution, resulting in a substantially

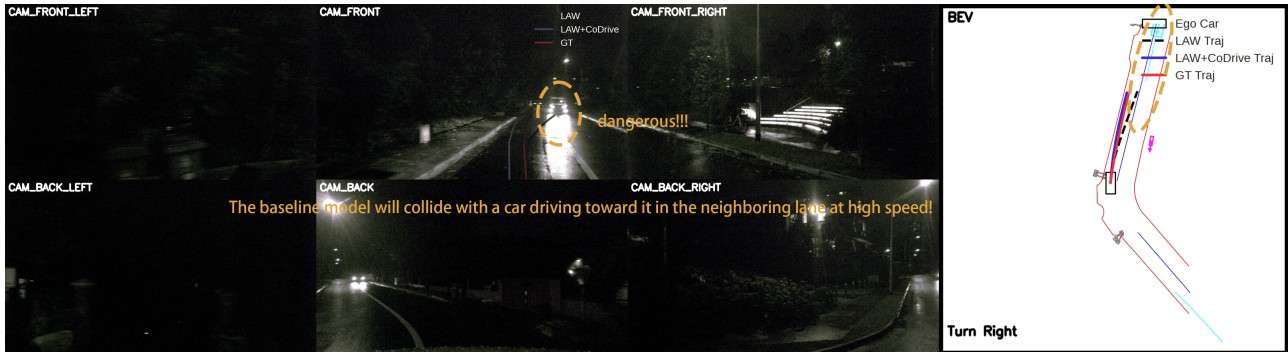

*Figure 7.* The baseline model collides with a fast-approaching vehicle in the adjacent lane, whereas our model successfully avoids the collision.

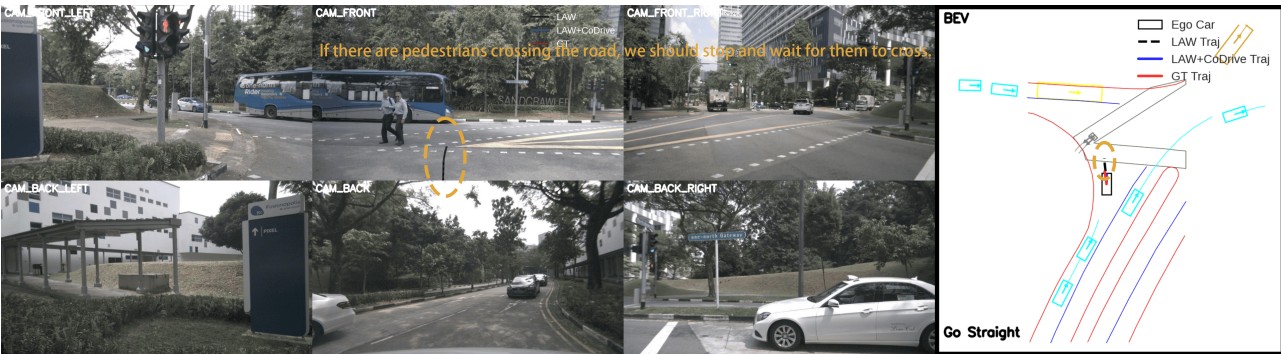

*Figure 8.* When pedestrians are crossing ahead, our model stops and yields, while the baseline model continues moving forward.

larger L2 distance. When the competitive mechanism is introduced, it suppresses value overestimation and pulls the RL actor back toward the expert policy, enabling exploration within a safer trust region.

### 5.2. Qualitative Analysis

We further visualize the planned trajectories on nuScenes to qualitatively compare our method with the baseline LAW. The results show that, with reinforcement learning and self-exploration, our model is better at recognizing hazardous situations and proactively avoiding potential collisions with surrounding vehicles and pedestrians. In contrast, the baseline LAW often fails to react appropriately, likely due to its reliance on imitating expert trajectories without understanding the underlying intent (see Fig. 7 and Fig. 8). Additional qualitative examples are provided in Appendix G.

## 6. Conclusion

We present CoIRL-AD, a competitive dual-policy framework that integrates imitation learning and reinforcement learning for end-to-end autonomous driving. To address the limited generalization and long-tail performance of pure imitation learning in offline settings, we leverage reinforcement learning as a complementary signal. We choose to conduct

IL and RL at the same stage instead of the normal two stage way as the RL in offline expert datasets are unstable, and we also decoupling IL and RL into separate actors to avoid gradient conflict when jointly optimize both IL and RL loss. A competition-based mechanism enables stable interaction and effective knowledge sharing between the two policies, while a latent world model supports long-horizon reasoning beyond immediate rewards. Experiments on the nuScenes dataset demonstrate consistent improvements over strong baselines, with particularly significant gains in cross-city generalization and long-tail scenarios. These results suggest that reinforcement learning, when properly structured and integrated, can substantially enhance offline end-to-end driving systems. We believe this work offers a promising direction for combining imitation and reinforcement learning in autonomous driving and broader embodied AI applications.

## Impact Statement

This work aims to advance the field of end-to-end autonomous driving by proposing a framework that integrates imitation learning and reinforcement learning into offline training without relying on external simulators. The techniques proposed are general-purpose and may find application in broader embodied AI settings.

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

## A. Step-aware Reinforcement Learning

When the RL actor explores, it samples an action sequence from $n$ policies (see Eq. 12). Since each action is modeled separately, naively sampling a full sequence draws actions independently from different Gaussian distributions. When the sampled sequence $\tau_a^{(g)}$ is accumulated into the position sequence $\tau_{pos}^{(g)}$, the resulting trajectory can become unstable or unsmooth, and may violate basic kinematic regularities. We visualize the comparison between naive group sampling and our step-aware method in Fig. 9 (straight driving), Fig. 10 (left turn), and Fig. 11 (right turn). The goal is to let the RL actor explore reasonable driving trajectories; directly exploring many unrealistic trajectories is inefficient, so we adopt a step-aware sampling mechanism.

More specifically, group sampling produces $G$ samples for each action in the action sequence. We decouple exploration by step: instead of sampling all $n$ actions simultaneously, each exploration samples only one action while setting the remaining $n-1$ actions to the mode of their corresponding policies, i.e., the Gaussian expectation $E[\pi_i]$. The process is visualized below and formalized in Algorithm 1.

---

**Algorithm 1** Step Aware RL with Group Sampling

---

**Input:** $\{\pi_1, \pi_2, ..., \pi_n\}$, $s$, $s_w$, $V_\psi$ (Critic Model), $(i, g, L_{actor}, L_{critic} \leftarrow 0)$

1: **repeat**
2:    $i \leftarrow i + 1$
3:    **repeat**
4:       $g \leftarrow g + 1$
5:       $\tau_a^{(g)} \leftarrow \{E[\pi_1], ..., a_i^{(g)} \sim \pi_i(a_i^{(g)}|s_{w,j\geq i}), ..., E[\pi_n]\}$
6:       Calculating reward $\tau_r^{(g)}$ based on Eq.8, 9
7:       Predict future state $s'^{(g)}$ based on Eq. 14
8:       Computing "long-term" advantage $A_{long}^{(g)}$ based on Eq. 15
9:    **until** $g = G$
10:    Computing critic advantage for step i, $A_{critic} = \text{Z-Score-Norm}(\{A_{long}^{(1)}, ..., A_{long}^{(G)}\})$
11:    $L_{actor} \leftarrow L_{actor} - \frac{1}{G}\sum_{g=1}^{G} A_{critic}^{(g)} \cdot \left(\sum_{j=1}^{n} \log^{\pi_j(\tau_a^g[j]|s_{w,k\geq j})}\right)$
12:    $L_{critic} \leftarrow L_{critic} + \frac{1}{G}\sum_{g=1}^{G}\left[V_\psi(s) - \left(\sum \tau_r^{(g)} + \gamma \cdot V_\psi(s'^{\hat{(g)}}))\right)\right]^2$
13: **until** $i = n$
14: $L_{actor} \leftarrow \frac{1}{n} \cdot L_{actor}$
15: $L_{critic} \leftarrow \frac{1}{n} \cdot L_{critic}$

**Output:** Loss of actor $L_{actor}$ and dreaming critic $L_{critic}$

---

## B. Benchmarks

**nuScenes** (Caesar et al., 2020) is a large-scale autonomous driving benchmark featuring 1,000 20-second urban driving scenes with 1.4M annotated 3D boxes across 23 object classes. It provides 360° imagery from six cameras and 2Hz keyframe annotations. Following prior works (Hu et al., 2023; Jiang et al., 2023), we evaluate planning using L2 placement error and Collision Rate.

## C. Implementation Details

For experiments on nuScenes, our method builds upon the LAW framework (Li et al., 2025b). We train the models using 8 NVIDIA A800-SXM4-80GB GPUs and perform evaluation on the same hardware. Training uses a batch size of 1, the AdamW optimizer, and a learning rate of $5 \times 10^{-5}$ with cosine annealing and linear warmup. All other training settings follow the original LAW and SSR configurations. The training process takes approximately 20 hours to complete.

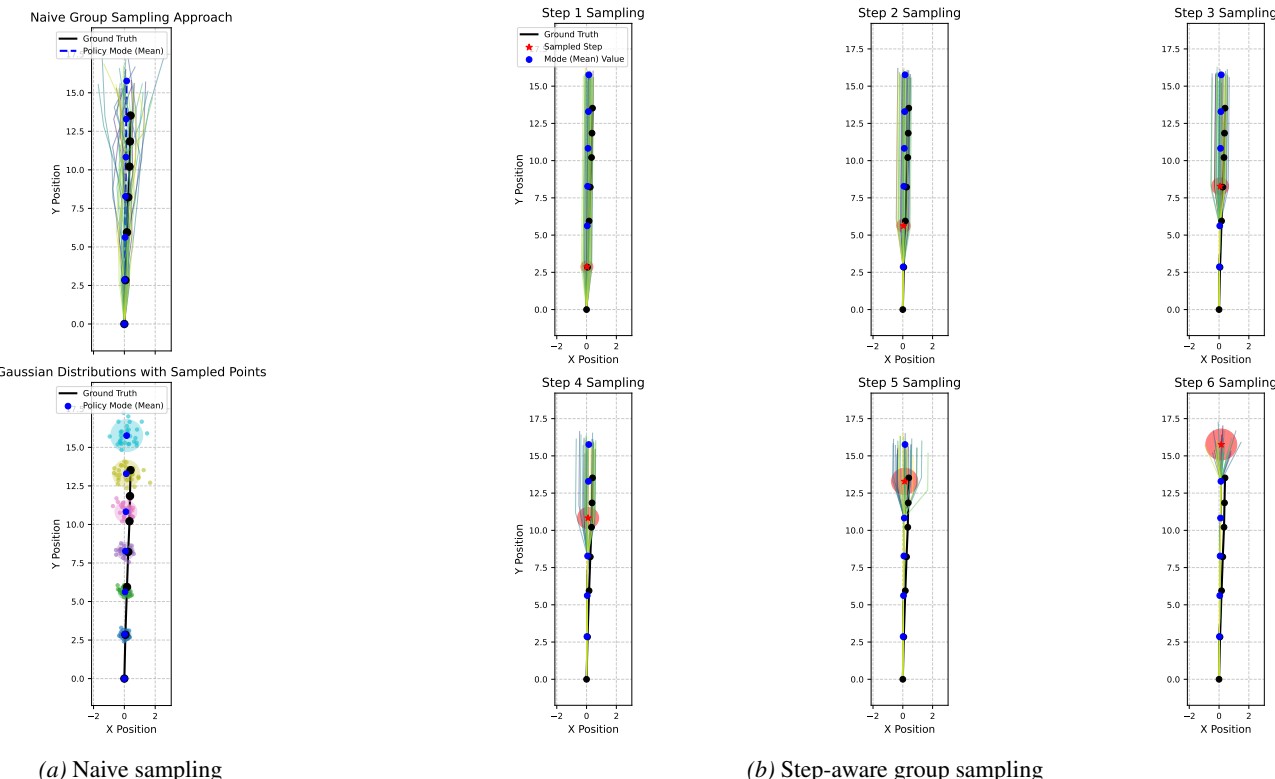

*(a)* Naive sampling        *(b)* Step-aware group sampling

*Figure 9.* Comparison between naive group sampling and step-aware group sampling in straight-driving scenarios during training.

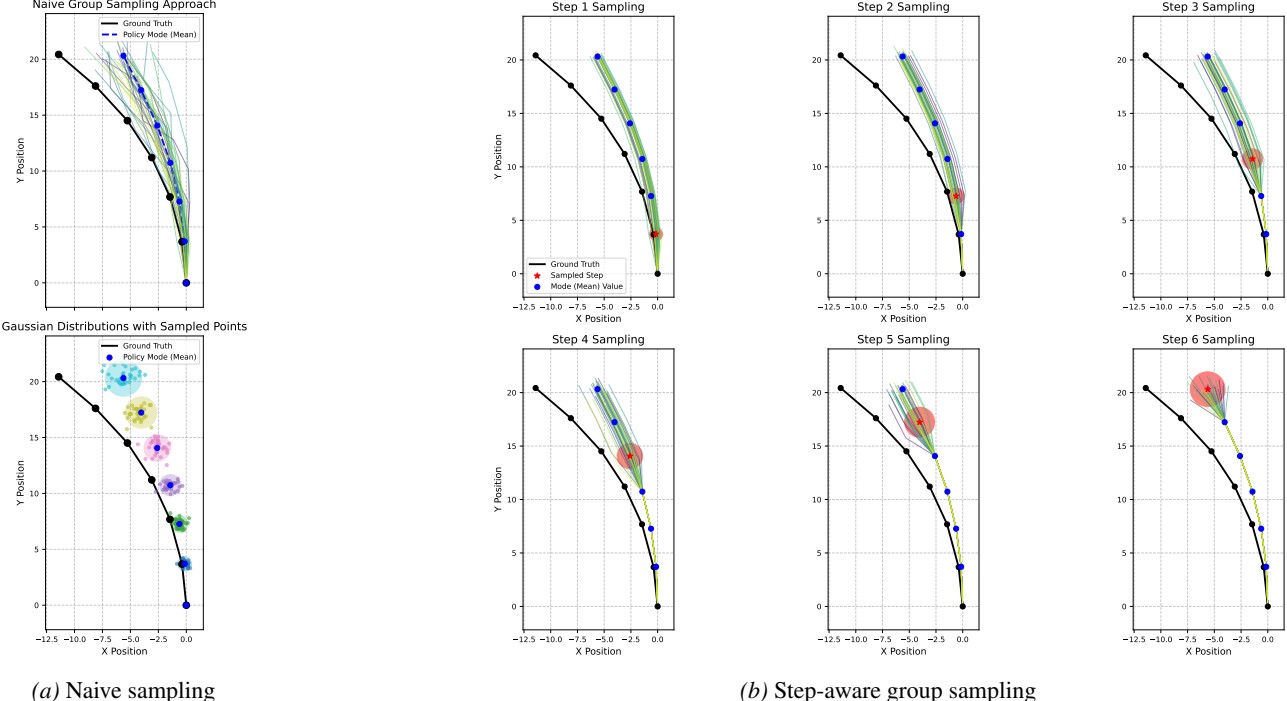

*(a)* Naive sampling        *(b)* Step-aware group sampling

*Figure 10.* Comparison between naive group sampling and step-aware group sampling in left-turn scenarios during training.

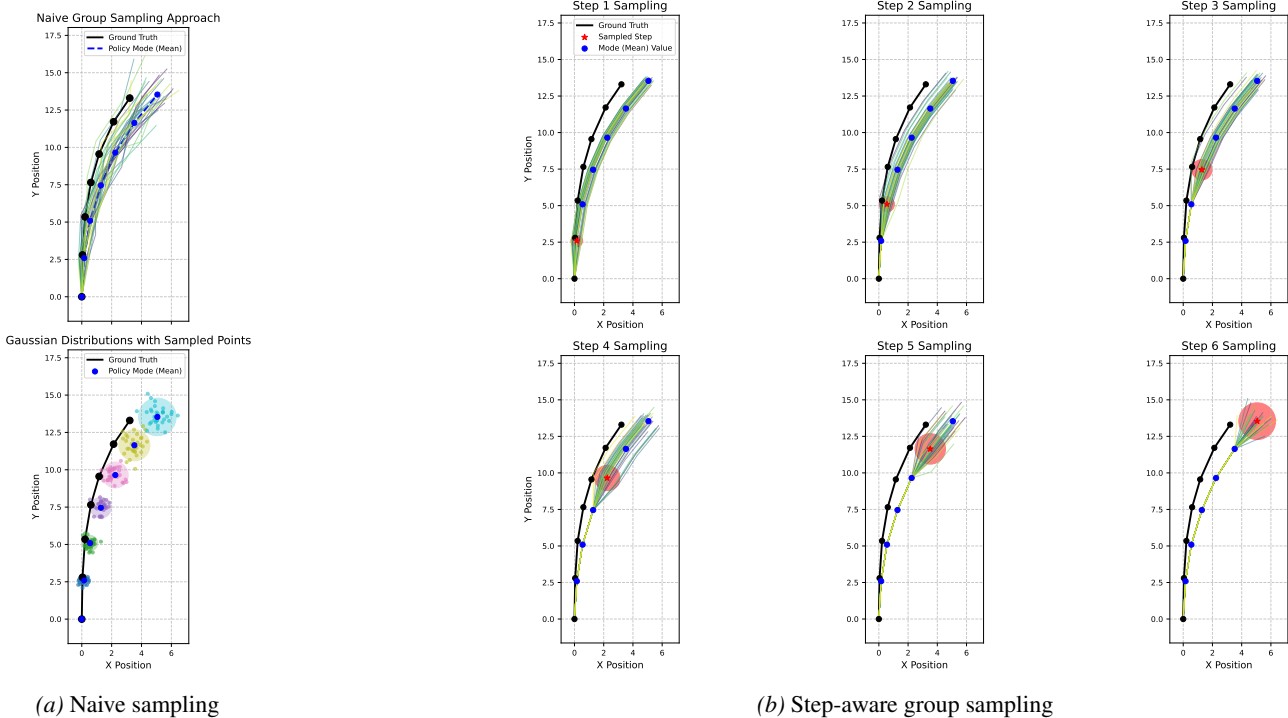

*(a)* Naive sampling

*(b)* Step-aware group sampling

*Figure 11.* Comparison between naive group sampling and step-aware group sampling in right-turn scenarios during training.

## D. Hyperparameters

We summarize all hyperparameters used in our experiments in Tab. 5. Since the world model is trained purely on an offline dataset, its predictions may become inaccurate when the RL actor explores states that deviate from the expert distribution. To mitigate the error caused by imagination of the world model, we adopt a smaller discount factor than commonly used in online RL methods (e.g., PPO (Schulman et al., 2017) with $\gamma = 0.99$). This choice places more emphasis on short-horizon rewards and stabilizes value learning under imperfect world model predictions.

*Table 5.* CoIRL-AD hyperparameters used in the nuScenes benchmark.

| Hyperparameter | Value |
|---|---|
| Interval between two competition ($k$) | 100 |
| Weight of world model loss ($\alpha$) | 0.2 |
| Discount factor ($\gamma$) | 0.5 |
| Weight of behavior cloning loss ($\beta$) | 0.005 |
| Competition minimum threshold ($\lambda_{\min}$) | 1 |
| Competition maximum threshold ($\lambda_{\max}$) | 10 |
| Soft linear interpolation parameter ($p$) | 0.5 |

Among all hyperparameters, $\beta$ plays a particularly important role, as it controls the contribution of the behavior cloning term $L_{bc}$ in the RL loss. Although the proposed competitive mechanism enables the RL actor to selectively learn from the IL actor (expert policy), the RL actor receives no direct expert guidance between two competition steps. This can lead to inefficient exploration, especially in the early stages of training.

To address this issue, we introduce a lightweight behavior cloning regularization that softly guides the RL actor toward expert-like behaviors while still preserving its ability to explore and optimize long-term returns. Importantly, this term is not intended to dominate the RL objective; instead, it serves as a stabilizing prior. Therefore, we restrict $\beta$ to small values.

The ablation study on $\beta$ is reported in Tab. 6. We observe that a moderate value ($\beta = 0.005$) achieves the best trade-off between trajectory accuracy and safety, while excessively large values (e.g., LAW, $\beta = \infty$) reduce the method to pure

imitation learning and degrade overall performance.

*Table 6.* Ablation study on the behavior cloning weight $\beta$.

| $\beta$ | L2 (avg) ↓ | Col (%) ↓ | L2 · Col ↓ |
|---|---|---|---|
| 0 | 0.65 | 0.20 | 0.13 |
| 0.00001 | 0.65 | 0.23 | 0.15 |
| 0.0001 | 0.70 | 0.20 | 0.14 |
| 0.0005 | 0.71 | 0.22 | 0.16 |
| 0.001 | 0.67 | 0.23 | 0.15 |
| **0.005** | **0.63** | **0.18** | **0.11** |
| 1 | 0.66 | 0.23 | 0.15 |
| LAW ($\beta = \infty$) | 0.66 | 0.22 | 0.15 |

Overall, these results indicate that a small amount of imitation regularization can significantly improve training stability without sacrificing the benefits of reinforcement learning.

## E. More Experiment Results

### E.1. Training Time

Our method introduces additional training cost due to the use of reinforcement learning, the RL actor's exploration, and the proposed competitive mechanism. The training time comparison is provided in Table 7. Although the training time roughly doubles, we consider this overhead acceptable. Most RL-based methods require long training schedules to converge, and in our case, the additional cost leads to a substantial improvement in generalization while leaving inference latency unchanged—a crucial property for embodied tasks such as autonomous driving.

*Table 7.* Training time on the nuScenes training set.

| Method | Training Epochs | GPU Usage | Time |
|---|---|---|---|
| LAW | 24 | 8×A800-80G | 10 h |
| CoIRL-AD | 24 | 8×A800-80G | 20 h |

### E.2. Inference Time

At inference time, our method introduces no additional latency. The knowledge learned by the RL actor has been transferred to the IL actor through competition, so only the standard forward pass is required during deployment. We test the inference speed of LAW and CoIRL-AD on a single A100 GPU, and the results are shown in Tab. 8.

*Table 8.* Inference speed and latency.

| Method | fps | latency (ms) |
|---|---|---|
| LAW | 26.99 | 37.1 |
| CoIRL-AD | 27.1 | 37.0 |

### E.3. Generalization and Performance on Long-tail Scenarios

**Details on Long-tail Subset Construction**   To avoid potential selection bias when only consider the performance of baseline model, we reconstructed the evaluation subsets using a strictly symmetric protocol. We identified hard scenarios from both CoIRL-AD and the LAW baseline using identical criteria, and then evaluated both methods on the union of these scenarios. This prevents the subset definition from favoring either method. The L2 Long-Tail Dataset is built by first selecting scenes with `fut_valid_flag=TRUE`, and then filtering for scenes with L2 distance greater than 0.3 at 1s, greater than 0.6 at 2s, and simultaneously greater than 1.0 at 3s of both baseline and our model, then union the two subset. This results in a total of 2247 scenes for testing. The Collision Rate Long-Tail Dataset is obtained by selecting scenes with

`fut_valid_flag=TRUE` and excluding all scenes with a zero collision rate at the 3s horizon of both baseline and our model, then union both subset, yielding 124 test scenes.

**Detailed Results**    We visualize the metrics of the baseline and CoIRL-AD in Fig. 12 and Fig. 3. Detailed results on the two long-tail subsets are shown in Tab. 9 and Tab. 10.

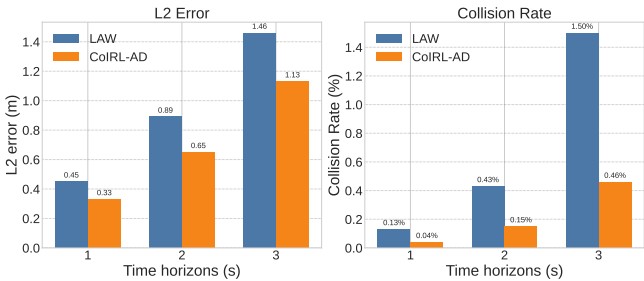

*Figure 12.* Cross-city Generalization Ability Visualization

*Table 9.* Long-tail dataset (L2) comparison results

| Method | L2 (m) ↓ | | | | Col (%) ↓ | | | | L2 · Col ↓ |
|---|---|---|---|---|---|---|---|---|---|
| | 1s | 2s | 3s | Avg. | 1s | 2s | 3s | Avg. | Avg. |
| LAW | 0.52 | 1.01 | 1.60 | 1.04 | 0.11 | 0.18 | 0.74 | 0.34 | 0.36 |
| CoIRL-AD | 0.45 | 0.92 | 1.53 | **0.97** | 0.07 | 0.14 | 0.53 | **0.25** | **0.24** |

*Table 10.* Long-tail dataset (Collision) comparison results

| Method | L2 (m) ↓ | | | | Col (%) ↓ | | | | L2 · Col ↓ |
|---|---|---|---|---|---|---|---|---|---|
| | 1s | 2s | 3s | Avg. | 1s | 2s | 3s | Avg. | Avg. |
| LAW | 0.40 | 0.94 | 1.67 | 1.01 | 3.63 | 5.04 | 18.82 | 9.16 | 9.22 |
| CoIRL-AD | 0.37 | 0.84 | 1.51 | **0.91** | 2.42 | 4.23 | 15.19 | **7.28** | **6.60** |

## F. Comparison with Other RL Methods

To validate our choice of a model-based RL optimization strategy, we compared our approach against traditional offline RL and alternative on-policy optimizers. The configurations are:

- **LAW (No RL)**: The baseline pure imitation learning model.

- **CoIRL (Proposed)**: Our model-based RL framework utilizing the world model.

- **CQL (Pure Offline RL)**: A standard conservative offline RL baseline. It only uses data from the offline dataset and does not utilize future states predicted by the world model.

- **PPO-lite (Model-Based "Online RL")**: Uses the offline dataset but employs the world model as a fast simulator. We implement PPO as an "on-policy" RL algorithm operating within this learned simulator (without a replay buffer).

The results in Tab. 11 reveal a severe performance gap between pure offline RL and our world-model-based approach. CQL fails in this setting, with L2@avg rising to 4.67. We attribute this to a data-regime mismatch: offline driving datasets are imitation-oriented and dominated by positive expert trajectories, lacking the negative or counterfactual state coverage required for stable value estimation in pure offline RL.

In contrast, using the world model as a simulator, as in CoIRL and PPO-lite, enables on-policy exploration of both positive and negative trajectories, yielding more stable and effective value learning. CoIRL and PPO-lite achieve similarly strong overall results with slight trade-offs: CoIRL tracks the expert better in terms of L2 error, while PPO-lite slightly reduces collisions. These results indicate that the framework is robust to the specific choice of optimizer and support our core claim that world-model-enabled exploration is critical in this offline setting.

*Table 11.* **Performance on nuScenes using different RL methods.**

| Description | L2 (m) ↓ | | | | Collision Rate (%) ↓ | | | | L2 · Col ↓ |
|---|---|---|---|---|---|---|---|---|---|
| | 1s | 2s | 3s | Avg. | 1s | 2s | 3s | Avg. | Avg. |
| LAW (No RL) | 0.32 | 0.63 | 1.03 | 0.66 | 0.09 | 0.12 | 0.46 | 0.22 | 0.15 |
| CoIRL (Proposed) | 0.29 | 0.59 | 1.00 | 0.63 | 0.06 | 0.10 | 0.37 | 0.18 | 0.11 |
| CQL (Pure Offline) | 2.80 | 4.67 | 6.55 | 4.67 | 1.76 | 3.56 | 4.88 | 3.40 | 15.87 |
| PPO-lite (Model-Based) | 0.32 | 0.62 | 1.03 | 0.66 | 0.02 | 0.08 | 0.37 | 0.16 | 0.10 |

# G. More Qualitative Results

In this section, we provide additional qualitative results to further demonstrate the effectiveness of our approach.

### G.1. Good Cases

The qualitative results show that, after incorporating reinforcement learning, our model avoids several collisions that the baseline fails to handle. This further supports the effectiveness of integrating IL and RL.

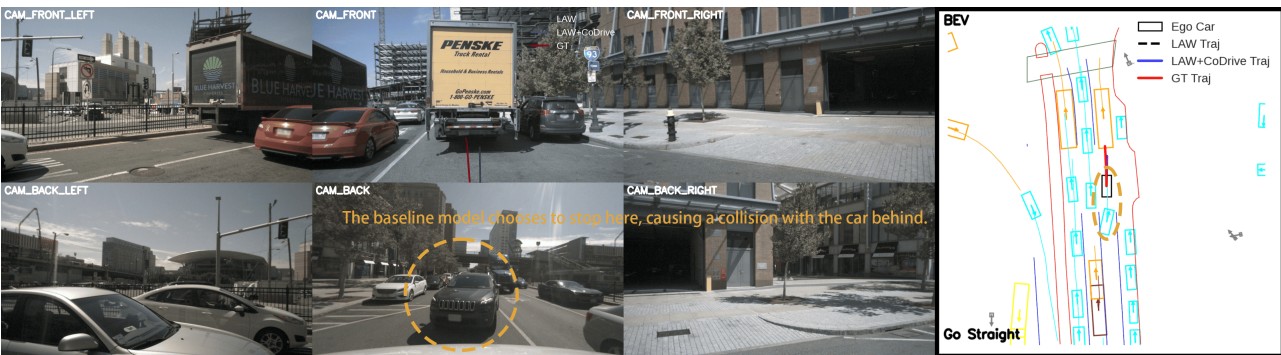

*Figure 13.* Good Case: In a queuing scenario while waiting for the green light, our model proceeds after the car ahead turns off its indicator and starts moving, whereas the baseline model remains stopped, which may lead to a rear-end collision.

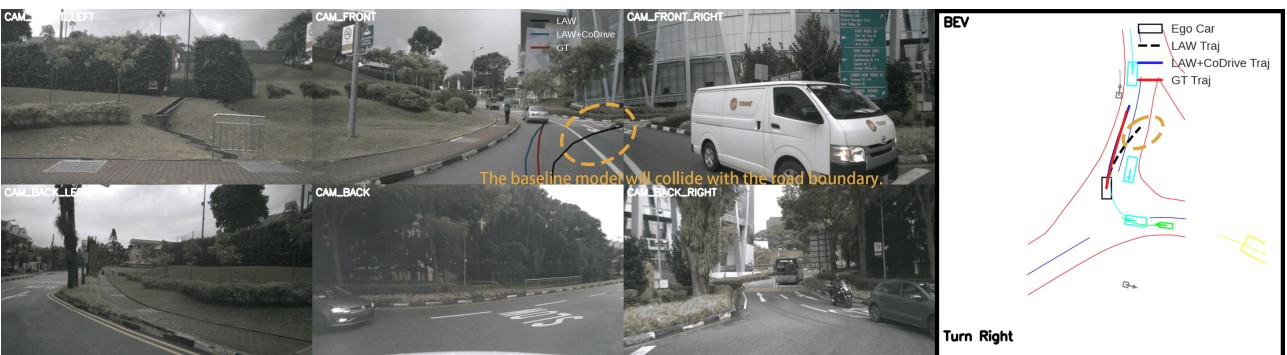

*Figure 14.* Good Case: When the high-level command is incorrect, our model adapts its planned trajectory based on the driving scenario, whereas the baseline model fails to adjust.

### G.2. Bad Cases

However, there are also some scenarios where both the baseline model and our model perform unsatisfactorily.

# H. Usage of LLMs

During our research, we used LLMs in the following ways:

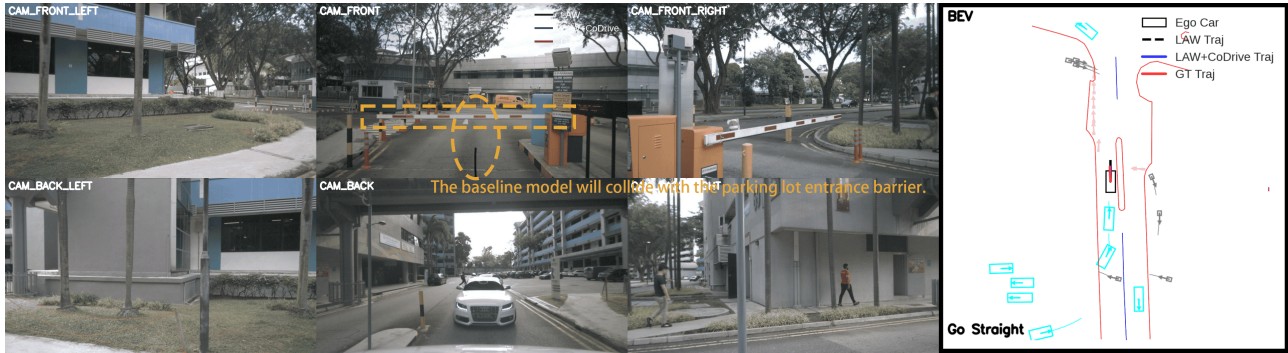

*Figure 15.* Good Case: When exiting the parking lot, our model waits for the toll barrier to rise, whereas the baseline model proceeds forward and collides with it.

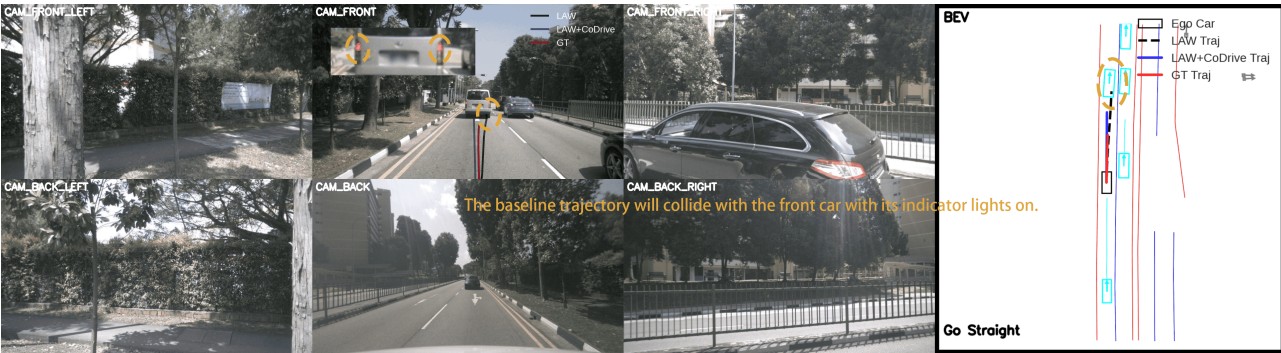

*Figure 16.* Good Case: When the leading vehicle has its indicator on, our model slows down to avoid a rear-end collision, whereas the baseline model maintains speed and causes a collision.

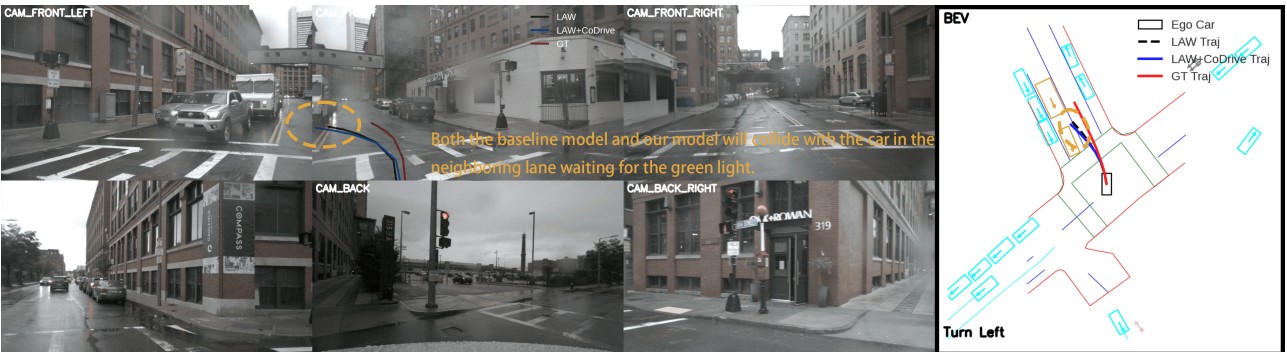

*Figure 17.* Bad Case: When turning right at an intersection, both the baseline model and our model plan trajectories that collide with a vehicle waiting for the green light.

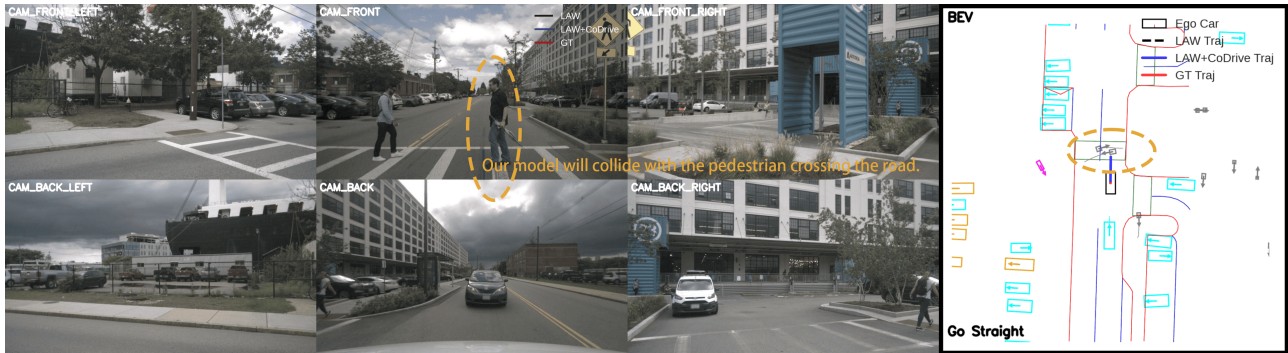

*Figure 18.* Bad Case: When pedestrians are crossing, our model continues to move forward at a low speed instead of stopping completely.

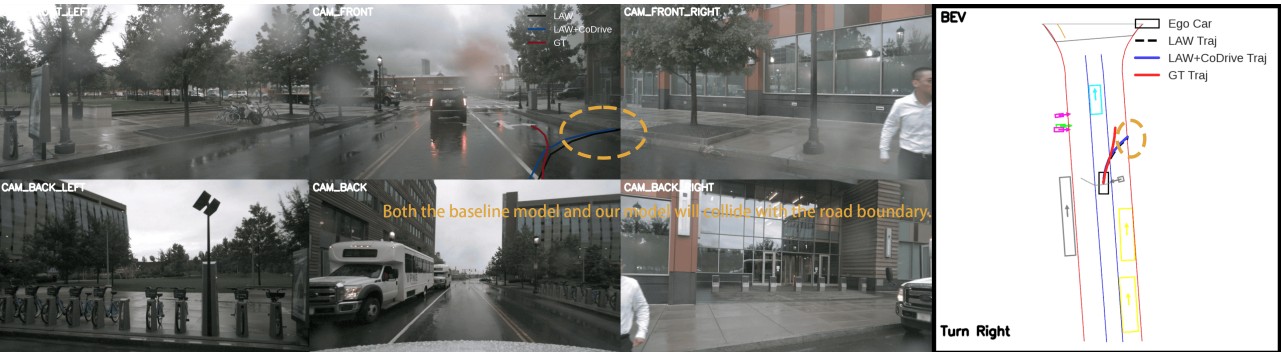

*Figure 19.* Bad Case: Both the baseline model and our model fail to plan reasonable trajectories in lane-changing scenarios.

- **Text polishing.** We employed LLMs to refine the writing style of our drafts and to shorten sections when the main text exceeded the page limit (9 pages). We did not use LLMs to generate new content; all polished text was manually reviewed before inclusion in the submission.

- **Idea exploration.** We used LLMs as a tool for brainstorming and literature search assistance. By discussing our ideas with LLMs, we were directed toward relevant research areas and key related works, which we then examined ourselves.

- **Code assistance.** LLMs were used to help debug programs by analyzing error logs, to review our code for potential issues, and to generate auxiliary visualization scripts. Except for visualization, we did not rely on LLMs to produce experimental code. For visualization, we first drafted data-loading code ourselves, then refined the visualization with LLM-generated snippets, carefully verifying correctness before use.

