# OpenReview forum: "CoIRL-AD: Collaborative-Competitive Imitation-Reinforcement Learning in Latent World Models for Autonomous Driving"
_ICML.cc/2026/Conference — ICML 2026 regular_

### Official Review · Reviewer_Noyd · 2026-02-24

**Soundness:** 3
**Presentation:** 3
**Significance:** 3
**Originality:** 3
**Overall Recommendation:** 4
**Confidence:** 4

**Summary:**

This paper proposes CoIRL-AD, a dual-policy framework that integrates IL and RL for offline end-to-end autonomous driving without external simulators. To address objective conflicts between IL and RL and instability in expert-only offline data, the method decouples the two into separate actors that share perception and a latent world model, while introducing a competitive mechanism to stabilize training through adaptive parameter exchange. The RL actor performs group-based trajectory sampling and evaluates imagined rollouts via the learned world model. Experiments on nuScenes show consistent improvements over strong baselines, particularly in cross-city generalization and long-tail scenarios, suggesting that structured IL–RL integration can improve robustness in offline driving systems.

**Compliance With Llm Reviewing Policy:**

Affirmed.

**Final Justification:**

This is a solid paper with clear practical value, and the rebuttal made the method and experiments more convincing. The main strengths are in the overall system design and empirical results, especially on harder generalization settings. I still think the novelty is more incremental than deep, so I see it as a weak accept rather than a stronger accept.

**Key Questions For Authors:**

1. Can the authors provide a more principled explanation of why the proposed competitive parameter swapping/merging mechanism stabilizes offline RL training? For example, is there analysis (theoretical or empirical) from the perspective of value overestimation, distribution shift, or optimization dynamics that supports its effectiveness?
2. Beyond final performance metrics, can the authors provide diagnostic evidence (e.g., evolution of value estimates, policy divergence between IL and RL actors, or world model prediction error over training) to better illustrate how the competitive mechanism mitigates instability?
3. Have the authors considered benchmarking against more advanced conservative or state-of-the-art offline RL methods (e.g., CQL-style approaches)? The current evaluation mainly compares different IL–RL integration strategies, leaving open questions about how competitive the method is within the broader offline RL literature.
4. Do the authors have evidence that the proposed framework generalizes to other offline driving datasets or alternative reward designs, beyond nuScenes? This would help clarify the broader applicability of the approach.

**Limitations:**

1. Limited theoretical justification of the competitive mechanism. The paper does not provide a principled analysis of why the proposed parameter swapping/merging strategy stabilizes offline RL, and the underlying optimization or value estimation dynamics remain unclear.
2. The evaluation primarily contrasts different IL–RL integration strategies, with limited benchmarking against stronger conservative or state-of-the-art offline RL methods, leaving questions about broader competitiveness.
3. Empirical validation is centered on nuScenes, with unclear generality across datasets or reward designs.
4. Additionally, minor grammar errors and outdated citation formatting (e.g., referring to published works as preprints) reduce the overall polish of the manuscript.
5. Missing Impact Statement.

**Strengths And Weaknesses:**

Strengths:
1. The paper addresses IL–RL integration in a strictly offline end-to-end autonomous driving setup without external simulators, which is both realistic and technically challenging.
2. Decoupling IL and RL into separate actors with a competitive interaction mechanism provides a clean structural solution to gradient conflicts and instability observed in naive joint or two-stage approaches.
3. The paper includes thorough ablations and demonstrates especially strong improvements in cross-city generalization and long-tail scenarios, supporting the claim that RL contributes most under distribution shift.

Weaknesses:
1. The main contributions are architectural and training-strategy based, with relatively limited theoretical insight into why the competitive mechanism ensures stability.
2. The evaluation does not extensively benchmark against stronger conservative or state-of-the-art offline RL approaches, leaving open questions about general applicability.
3. The manuscript contains several small grammar mistakes (e.g., subject–verb agreement errors such as “the results is shown”) and occasional awkward phrasing, which slightly affect readability. In addition, some cited works are still referred to as arXiv preprints even though they have been formally published, suggesting that the bibliography has not been fully updated. Missing Impact Statement.

---

> ### Author Rebuttal · Authors · 2026-03-31
>
> ### W1&Q1&Q2&L1 (and MQNy's Q3): Stability and Convergence of the Competitive Mechanism
> For the requested diagnostic evidence (evolution of value estimates, policy divergence, and mitigation of instability), please refer directly to **Exp1** in our response to Reviewer LjSX.
>
> Regarding theoretical vs. empirical convergence: While providing strict theoretical convergence guarantees for complex deep RL in continuous, high-dimensional spaces is notoriously challenging, our framework is built on principled architectural safeguards that ensure highly robust empirical convergence:
>
> 1. **Built-in Stabilization:** As shown in **Exp1**, the Competitive Mechanism acts as a structural regularizer. By utilizing the IL actor as a stabilizing pivot, we safely constrain RL exploration to a "trust zone," mathematically preventing the RL actor from exploiting over-estimated values caused by world model OOD hallucinations.
> 2. **Hyperparameter Robustness:** Our empirical convergence is not brittle. For example, our ablations demonstrate that varying the weight parameter $\beta$ does not cause training collapse or sharp metric degradation (see Table 6 in the main paper).
>
> Summary: We do not claim strict theoretical bounds; rather, our core contribution is a mechanism design that ensures robust, empirically verifiable convergence across multiple datasets (see **Exp3** in our response to Reviewer LjSX) and hyperparameter configurations.
>
>
> ### W2&Q3&L2: Ablation Other RL methods
> We appreciate this suggestion. To justify our algorithmic design, we explicitly benchmarked against a strong offline RL baseline (CQL). Please see **Exp4** below. The results reveal a critical insight: pure offline RL catastrophically fails in our setting because offline driving datasets are dominated by safe expert demonstrations and lack the negative counterfactuals required for stable offline value estimation. This fundamentally validates our use of a latent world model as a simulator.
>
>
>
> ### W3&L4&L5: Manuscript Polish, Bibliography, and Impact Statement
> We sincerely thank the reviewers for their attention to detail. We commit to addressing all structural and formatting omissions in the camera-ready revision:
> - **Concept Flow:** We will add a "Core Concepts and Paper Organization" paragraph at the end of the Introduction to preview key design choices early on.
> - **Grammar & Legibility:** We apologize for the grammatical errors and awkward phrasing. We will conduct a rigorous, line-by-line proofreading pass to elevate readability.
> - **Bibliography:** We will update our references to replace arXiv preprints with their formal publication venues.
> - **Impact Statement:** We will add a dedicated section discussing the broader societal impacts and ethical considerations of our framework.
>
>
>
> ### Q4&L3: Generalizes to Other Offline Driving Datasets/Alternative Reward Designs,
> Please refer to **Exp3** in our response to Reviewer LjSX. We successfully adapted CoIRL-AD to the entirely separate NAVSIM benchmark. Crucially, in this experiment, we utilized components of the official PDM score as our RL reward function (which differs significantly from the nuScenes reward design). This definitively proves our framework generalizes across both different offline datasets and alternative reward structures.
>
> ---
>
> ### Exp4 (for W2&Q3&L2): Ablations of Different RL Methods, Offline RL v.s. (fake) Online RL in Our Setting
>
> To justify our algorithmic design, we compared CoIRL against pure offline RL (CQL, ignoring the world model) and an alternative model-based optimizer (PPO-lite, using the world model as a simulator).
> - **The Failure of Pure Offline RL:** CQL catastrophically collapsed (L2@avg spiked to 4.67 vs CoIRL's 0.63). Offline driving datasets are dominated by positive expert demonstrations, lacking the negative counterfactuals needed for stable offline value estimation.
> - **The Necessity of the Simulator:** Model-based optimization (CoIRL and PPO-lite) solves this. By using the latent world model as a simulator, the agent explores negative trajectories "on-policy," enabling stable value learning.
> - **Robustness:** CoIRL and PPO-lite achieve similar strong performance (CoIRL optimizes L2 slightly better; PPO-lite optimizes collision slightly better). This proves our framework's success stems from the *world-model-enabled exploration*, rather than a hyperspecific optimizer.
>
> For full results, baseline comparisons, and logs, see [exp4 **(click here)**](https://anonymous.4open.science/r/drive-with-two-minds/icml_rebuttal/exp4/README.md) in our anonymous repo.

---

> > ### Author Rebuttal · Reviewer_Noyd · 2026-04-01
> >
> > Thank you for the clarifications and the additional pointers. I appreciate the authors’ detailed response and the effort to address the concerns raised by the reviewers. The rebuttal is helpful for better understanding the motivation and empirical scope of the work.

---

> > > ### Author Response · Authors · 2026-04-05
> > >
> > > Dear Reviewer Noyd,
> > >
> > > Thank you for your continued support and for confirming that your concerns have been fully resolved. We truly appreciate the time you took to deeply engage with our work.
> > >
> > > Your initial questions regarding the underlying optimization dynamics and the comparison to pure offline RL prompted us to run ablations (like our CQL comparison) that significantly strengthened the paper's scientific narrative. We will rigorously incorporate these clarifications, the new baseline comparisons, and the expanded impact statement into the final version of the manuscript.
> > >
> > > Thank you again for your valuable and meticulous feedback!

---

### Official Review · Reviewer_gMNL · 2026-03-04

**Soundness:** 2
**Presentation:** 2
**Significance:** 2
**Originality:** 3
**Overall Recommendation:** 2
**Confidence:** 4

**Summary:**

The paper presents a novel offline training method for autonomous driving that combines imitation and reinforcement learning with latent world models in a “competitive” dual policy framework. In the competitive framework two policies are trained simultaneously, and their parameters are periodically shared (soft-merged or replaced) depending on relative policy performance. Together with the policies the framework co-trains a latent world model that acts as a regularizer and is used for the offline-RL critic. Additionally, the method introduces an inverse-causal attention head for the policy that generates output waypoints in a “backward” manner.  Results for open-loop planning on the NuScenes dataset show improvement over baselines, especially in the cross-city generalization setting.

**Compliance With Llm Reviewing Policy:**

Affirmed.

**Final Justification:**

I believe this paper requires a major revision that would necessitate a thorough new review, so I maintain my negative assessment. That said, I encourage the authors to continue pursuing this line of work.

**Key Questions For Authors:**

Please comment on the key limitations summarized above.

**Limitations:**

The paper does not explicitly address its limitations or potential negative societal impacts.

**Strengths And Weaknesses:**

### Strengths:

* The paper addresses an important open problem in autonomous driving: training robust, human-like policies that generalize effectively to rare and long-tail scenarios.
* The general direction of fusing RL and IL components is interesting and represents an active area of research
* The paper has multiple interesting elements that have not been combined in the same way before (competitive framework, shared latent world model, inverse-causal attention).
* While in the nominal evaluation setting the gains are small, the cross-city generalization results look strong.

### Weaknesses:
#### Novelty is unclear.
* The paper presents a large number of ideas, but it is often unclear which constitute novel contributions, which build on prior work, and which amount to ad hoc training heuristics.
* The ideas that appear to be novel—such as weight sharing between the two policies—come across as somewhat heuristic. It would strengthen the paper to evaluate these components in domains beyond driving. The backward-planning attention mechanism is also likely novel; however, it seems more like a domain-specific design choice and is not presented as a central contribution.

#### The overall quality of the presentation falls below expectations.
* Some concepts are not communicated clearly, for example, the relatively simple idea behind what makes the framework “competitive” remains a mystery to the reader until Sect 3.4.
* While in the method section 3.2 “backward planning” is presented as an important design choice, it is not mentioned at all in any prior sections.
* Significant grammatical errors/typos make comprehension difficult at times (for example “we find that although ignore this term can also get better result, add this term with a small \beta can get even better results”).

#### Evaluation has issues.
* The strongest results are in the “Long-Tail Scenarios” evaluation (Fig. 3), however, this setting seems to be severely biased. The evaluation set is constructed by first running the baseline model and then selecting the examples on which it performs worst. Then the proposed method is compared to the same baseline on those examples. This setting appears to suffer from significant selection bias.
* Evaluation is open-loop only, it is unclear how the results would transfer to the more relevant closed-loop setting.
* The RL algorithm relies on a jointly learned latent world model, however, there is limited evaluation of the world model itself.

#### Gains cannot be clearly attributed to key contributions.
* The main results in Table 1 indicate that the gains over the most relevant baseline, LAW, are present but modest. However, Table 3 suggests that in fact the gain over LAW comes only from the “backwards planning” design which is not a core contribution (i.e. CoIRL-AD w. causal mask performs same or worse than LAW). While the generalization results show a larger gap with respect to LAW, there are no other methods in this comparison, and one of the evaluation settings is biased (see above).

Conclusion: The paper contains some interesting ideas; however, in its current form, presentation issues make them difficult to fully appreciate. Several key components come across as incremental heuristics, the reported gains are not clearly attributed to the central contributions, and the long-tail evaluation setting—where the largest improvements are observed—appears to be biased.

---

> ### Author Rebuttal · Authors · 2026-03-31
>
> ### W1: Clarification of Novelty and Core Contributions
> Our framework is a principled solution to the IL and RL integration bottleneck in autonomous driving, not a collection of ad-hoc heuristics.
>
> **1. Motivation and Core Novelty:** Pure IL suffers from compounding errors ([Imitation is not Enough]). While recent models ([LAW, DriveVLA-W0]) use latent world models strictly as passive decoders, **our core novelty** transforms them into *active, interactive simulators*. This enables safe *online* RL (akin to [GRPO]) within the latent space, overcoming offline exploration limits.
>
> **2. Distinction from Existing Paradigms:**
> - **vs. Pure Model-Based RL:** Unlike [DreamerV3, DreamerV4, Think2Drive] which train from scratch, we anchor the RL agent with offline expert demonstrations—crucial for industrial driving.
> - **vs. Standard IL+RL:** While methods like [RAD, RecogDrive, imitation is not enough] also utilize world models as simulators, our framework systematically resolves the two major bottlenecks of IL/RL integration: (1) destructive gradient conflicts from combined objectives, and (2) catastrophic forgetting caused by sequential two-stage training (IL then RL).
>
> **3. Theoretical Grounding:** Our components are deliberate adaptations:
> - **Backward Planning:** Scaled from robotics ([Efficient Robotic Policy Learning via Latent Space Backward Planning]) to driving.
> - **Decoupled Actors:** Adapted from general RL ([IN-RIL: Interleaved Reinforcement and Imitation Learning for Policy Fine-Tuning]) to mitigate gradient conflicts.
> - **Competitive Mechanism:** Inspired by [GAN] minimax optimization, it stabilizes knowledge-swapping and prevents RL actors from exploiting world-model hallucinations.
>
>
>
> ### W2: Presentation Quality and Manuscript Flow
> We agree that structural clarity is paramount. Please refer to our detailed revision plan in our response to Reviewer Noyd’s W3, L4 & L5 (Manuscript Structure, Polish)
>
>
>
> ### W3: Biased “Long-tail Scenarios” Evaluation & Close-loop Evaluation & World Model Evaluation
> - **Resolving Selection Bias:** To resolve potential long-tail selection bias, we conducted a rigorous symmetric evaluation (**Exp2** below), confirming robust gains on strictly unbiased datasets.
> - **Closed-Loop Evaluation:** To address open-loop concerns, we successfully deployed our framework in NAVSIM, a closed-loop, reactive environment (see **Exp3** to Reviewer LjSX), confirming its generalizability.
> - **World Model Evaluation:** Since our WM operates entirely in the latent space—optimized for *control* rather than pixel generation—metrics like PSNR are inapplicable. Instead, we validate its efficacy through its stable reconstruction loss convergence during training, and the final RL policy's significant downstream performance gains.
>
>
> ### W4: Attribution of Gains and Evaluation Bias
>
> We respectfully clarify the interpretation of our ablation studies, which demonstrate that our core contributions are highly interdependent. For evaluation bias concern, please see Exp2 in the bottom.
>
> - **Clarifying Table 3 (The Role of Backward Planning):** Table 3 does not imply that backward planning is the sole source of improvement over LAW. The LAW baseline does not utilize this attention module at all. Table 3 strictly ablates the *masking strategy* within the CoIRL framework, demonstrating that standard causal or unmasked attention degrades the RL policy, whereas inverse masking (backward planning) optimizes it.
> - **The Synergy of Contributions (Table 4):** The performance gains do not come from backward planning in isolation. As shown in Table 4 (line 6), utilizing backward planning without the competitive mechanism causes a significant performance drop. The gains stem strictly from the *combination* of decoupled actors, the competitive mechanism, and backward planning. *(For a detailed diagnostic proof of why the competitive mechanism is essential for stabilization, please refer to **Exp1** in our response to Reviewer LjSX).*
>
> ---
>
> ### Exp2 (for W3): Fair Long-Tail Evaluation Under Symmetric Subset Construction
>
> To rule out selection bias in our long-tail analysis, we constructed new subsets using a strictly symmetric protocol: evaluating on the union of hard cases from both CoIRL and the LAW baseline.
> - **Methodology:** We identified 2,247 Long-tail L2 scenes and 124 Long-tail Collision scenes using identical failure thresholds for both models.
> - **Results:** CoIRL consistently outperforms LAW on these unbiased subsets. For instance, on the Long-tail Collision subset, CoIRL reduces Col@3s from 18.82% to 15.19%, and L2@3s from 1.67 to 1.51 compared to LAW.
> - **Conclusion:** This confirms CoIRL's robustness in challenging, long-tail scenarios is genuine and not an artifact of evaluation construction.
>
> Full dataset breakdowns and results tables are available in [exp2 **(click here)**](https://anonymous.4open.science/r/drive-with-two-minds/icml_rebuttal/exp2/README.md) of our anonymous repository.

---

> > ### Author Rebuttal · Reviewer_gMNL · 2026-04-02
> >
> > I'd like to thank the authors for their rebuttal. I believe this paper requires a major revision that would necessitate a thorough new review, so I maintain my negative assessment. That said, I encourage the authors to continue pursuing this line of work.

---

> > > ### Author Response · Authors · 2026-04-05
> > >
> > > Dear Reviewer gMNL,
> > >
> > > Thank you for your continued engagement and for encouraging this line of work. We completely agree with your assessment that a clearer attribution of our performance gains (W4) is necessary for a major revision.
> > >
> > > You correctly noted that Table 3 and Table 4 in the original manuscript made it difficult to disentangle the contributions of the **Inverse Causal Mask (`inv.ar`)** and the **Competitive Mechanism**. To resolve this and directly address your hypothesis that the gains might stem solely from the masking design, we have run comprehensive new ablations during this discussion phase.
> > >
> > > We evaluated the components both in isolation and together. The new empirical data reveals a strict, necessary synergy between the two modules:
> > >
> > > | Method | Description | L2 $\cdot$ Col $\downarrow$ |
> > > | :--- | :--- | :--- |
> > > | **1. LAW (Baseline)** | Pure IL | **0.15** |
> > > | **2. LAW + inv.ar** | Pure IL + Mask | **0.18** *(Degrades)* |
> > > | **3. CoIRL-AD (w/o comp)** | RL + Mask (No Competition) | **0.17** *(Degrades)* |
> > > | **4. CoIRL-AD (Full)** | RL + Mask + Competition | **0.11** *(SOTA)* |
> > >
> > > **Analysis & Intuition:**
> > > 1. **The Mask is not an isolated heuristic (Row 1 vs Row 2):** Applying the inverse causal mask to a pure IL baseline actually *hurts* performance. Because IL relies on mimicking forward-time expert demonstrations, breaking forward causality confuses the agent.
> > > 2. **The Mask is an RL Exploration Engine:** Unlike IL, RL optimizes for future rewards. The `inv.ar` mask provides a vital inductive bias for goal-oriented search (backward planning). However, as shown in Row 3, allowing the RL agent to use this powerful engine *without* the competitive mechanism leads to destabilization and performance drops.
> > > 3. **The Core Contribution is Synergy:** The SOTA performance (0.11) is only unlocked when both components are active. The RL agent requires the mask to find trajectories superior to the expert's, while the entire framework requires the competitive mechanism to safely stabilize this exploration and prevent the RL actor from collapsing.
> > >
> > > *(Note: To keep this response concise, we have provided the summary matrix above. We also ran an exhaustive ablation testing various mismatched masking strategies between the IL and RL actors (e.g., IL using causal while RL uses inverse causal). These extended results confirm that unaligned masks break the representation sharing of the competitive mechanism, further validating our cohesive design. The full 7-row data table and analysis are available at [exp5 **(click here)**](https://anonymous.4open.science/r/drive-with-two-minds/icml_rebuttal/exp5/README.md)).*
> > >
> > > We believe this explicitly resolves the concerns regarding the attribution of gains (W4). This chronological ablation and the accompanying intuition will be integrated directly into the methodology and experimental sections of the camera-ready manuscript, satisfying the need for structural revision.
> > >
> > > Thank you again for pushing us to clarify this critical architectural synergy.

---

### Official Review · Reviewer_MQNy · 2026-03-11

**Soundness:** 3
**Presentation:** 3
**Significance:** 3
**Originality:** 2
**Overall Recommendation:** 4
**Confidence:** 4

**Summary:**

This paper presents CoIRL-AD, a competitive dual-policy framework that integrates IL and RL for offline end-to-end autonomous driving, implemented on the LAW latent world model framework. It decouples IL and RL actors to resolve objective conflicts, adopts backward planning for trajectory optimization and step-aware sampling for smooth RL exploration, and designs a competitive mechanism for adaptive knowledge sharing between actors. Evaluations on nuScenes show improved performance in collision reduction, cross-city generalization and long-tail scenarios against the LAW baseline. However, the work has notable limitations in cross-framework adaptability, result interpretation, convergence analysis and practical deployment, calling for further clarification and validation.

**Compliance With Llm Reviewing Policy:**

Affirmed.

**Final Justification:**

I appreciate the authors’ thorough response and supplementary experiments. The supplementary results help clarify framework generalization, result robustness, and training stability. The proposed decoupled IL-RL design is solid and meaningful for offline end-to-end autonomous driving.
Therefore, I change my score to 4.

**Key Questions For Authors:**

1. Can CoIRL-AD (or its core design ideas) be adapted to autonomous driving frameworks other than LAW, possibly with minor modifications? Are there any preliminary experimental results to support its cross-framework applicability?
2. How is the reliability of the experimental results ensured? Could the modest metric improvements be attributed to randomness, and what measures have been taken to exclude this possibility?
3. Can the convergence of CoIRL-AD be guaranteed theoretically or empirically? Does the framework achieve stable convergence across most training scenarios and different experimental settings?

**Limitations:**

Yes.

**Strengths And Weaknesses:**

Strengths:
1. The competitive dual-policy design skillfully decouples IL and RL to alleviate gradient conflicts, with adaptive knowledge sharing effectively anchoring RL exploration to expert IL behavior while retaining RL’s exploratory advantage for complex scenarios.
2. Tailored RL modules (step-aware sampling, dreaming critic) and innovative backward planning are well-suited to offline autonomous driving constraints, addressing key issues of unsmooth exploration and short-sighted planning in existing methods.
3. The study conducts thorough experiments on nuScenes, focusing on cross-city generalization and long-tail scenarios, which is core pain points of pure IL, with extensive ablation studies validating the efficacy of each key modular design.

Weaknesses:
The method is developed and validated on the LAW framework, with its core design explored primarily in this latent world model-based setup; cross-framework adaptation strategies and corresponding experimental verification are not provided, leaving its general applicability to other driving frameworks unexamined.
2. Overall performance improvements are relatively modest and may be subject to randomness. Notably, the 1s collision rate is slightly higher than the LAW baseline, yet the authors offer no analysis or explanation for this phenomenon and only emphasize aggregate average metrics.
3. The framework’s convergence and training stability lack clear and rigorous illustration, with no theoretical proofs or quantitative analyses to verify the reliability of the competitive dual-policy learning mechanism.
4. The method doubles the training cost compared to the baseline, a critical factor for real-world deployment that is underemphasized, with no lightweight optimization strategies proposed to mitigate this overhead.
5. Visual presentation can be refined: the legend positions in Figures 5 and 6 are not optimal, which impairs the visual clarity of the qualitative experimental results.

---

> ### Author Rebuttal · Authors · 2026-03-31
>
> ### W1&Q1: Cross-framework Adaptation
> Please refer to **Exp3** (in our response to Reviewer LjSX), where we successfully adapted CoIRL-AD’s core design to NAVSIM (a distinctly different world model-based framework). These closed-loop results confirm our method is framework-agnostic and broadly applicable.
>
>
> ### W2&Q2: Reliability of Improvements and Use of Aggregate Metrics
> - **Modest Margins & Avoiding Randomness:** The nuScenes dataset is heavily saturated with nominal, "easy" driving scenarios where the LAW baseline already excels, naturally diluting aggregate improvements. To confirm our gains are statistically significant and not random noise, we evaluated on highly challenging, out-of-distribution tasks where pure IL typically fails. CoIRL demonstrates undeniable, robust performance gaps in **Cross-City Generalization** (Table 2) and **Long-Tail Scenarios** (Table 9&10). *(Note: Please also see **Exp2** in our response to Reviewer gMNL for a rigorous, symmetric long-tail evaluation that eliminates selection bias).*
> - **The 1s Collision Rate Anomaly:** The slight increase at 1s is not random; it is an expected mathematical outcome of our RL formulation. As defined in Eq. 8 and 9, our objective minimizes the *cumulative* trajectory penalty. The RL agent learns a global policy that trades minor short-term deviations for critical long-term safety (e.g., accepting a marginal risk at $t=1s$ to avoid a severe crash at $t=3s$). Table 1's final column (L2@avg * Col@avg) confirms successful global optimization. For deployments requiring strict short-term safety, the reward function can simply be re-weighted to heavily penalize early collisions.
>
>
> ### W3&Q3: Convergence and Stability of the Competitive Mechanism
> - **Empirical Reliability:** Please refer to **Exp1** (in our response to Reviewer LjSX). Our diagnostic analysis quantitatively proves that the competitive mechanism successfully caps value over-estimation, which in turn strictly stabilizes RL training and prevents policy collapse.
> - **Theoretical Guarantee:** For a deeper theoretical and quantitative insight into how convergence is achieved, please see our response to Reviewer Noyd (**W1/Q1/Q2/L1**).
>
>
> ### W4: Training Cost vs. Real-World Deployment
> We agree that mitigating computational overhead is valuable, but we clarify the impact on deployment:
> - **Zero Inference Overhead:** For real-world deployment, the critical bottleneck is *inference latency*, not training time. As shown in Table 8, CoIRL's inference latency is **identical to the baseline**. The computationally heavy world model, critic, and RL actor are exclusively used during training and are entirely discarded during deployment.
> - **The Paradigm Trade-off:** A 2x training overhead is a reasonable and expected cost for a framework that unifies representation pre-training (IL) and alignment post-training (RL) into a single pipeline.
> - **Proposed Optimization Strategies:** To further reduce training costs for large-scale engineering, we propose two strategies: (1) **Learnable Reward Models:** Replacing complex rule-based functions with a lightweight model to eliminate CPU-bound bottlenecks during GPU training. (2) **Dynamic Rollouts:** Dynamically decaying the RL rollout group size as the policy stabilizes, trading marginal performance for significant speedups.
>
>
>
> ### W5: Visual Presentation can be Refined
> We completely agree. We will adjust the legend placements in Figures 5 and 6 and thoroughly review all visual elements for the camera-ready manuscript to ensure no data points or qualitative results are obscured.

---

> > ### Author Rebuttal · Reviewer_MQNy · 2026-04-02
> >
> > I appreciate the authors’ thorough response and supplementary experiments. The supplementary results help clarify framework generalization, result robustness, and training stability. The proposed decoupled IL-RL design is solid and meaningful for offline end-to-end autonomous driving.
> > Therefore, I change my score to 4.

---

> > > ### Author Response · Authors · 2026-04-05
> > >
> > > Dear Reviewer MQNy,
> > >
> > > Thank you so much for your positive feedback and for raising your score! We are thrilled that the supplementary experiments successfully clarified the framework's generalization, robustness, and training stability.
> > >
> > > We also deeply appreciate your recognition of our decoupled IL-RL design. Your initial questions pushed us to rigorously prove the robustness of this architecture, resulting in a much stronger paper. We will ensure that all the new experiments and optimizations discussed during the rebuttal phase are prominently integrated into the camera-ready manuscript.
> > >
> > > Thank you again for your time, effort, and highly constructive guidance!

---

### Official Review · Reviewer_LjSX · 2026-03-12

**Soundness:** 2
**Presentation:** 2
**Significance:** 3
**Originality:** 3
**Overall Recommendation:** 4
**Confidence:** 3

**Summary:**

This paper presents CoIRL-AD, a dual-actor training framework that integrates Imitation Learning (IL) and Reinforcement Learning (RL) for end-to-end autonomous driving. The primary motivation is to tackle the poor generalization of pure IL in long-tail scenarios while stabilizing RL training in an offline setting without external simulators.

On the methodology side, the IL component builds upon the LAW architecture but introduces an "inverse causal mask" for backward planning. For the RL part, the authors designed a training regime inspired by GRPO, leveraging a Latent World Model to simulate future states and provide reward signals without a physical simulator. These two actors are then fused through a "competitive mechanism" that adaptively updates parameters based on their relative performance.

Evaluation is conducted on the nuScenes dataset. The results show a noticeable reduction in collision rates, particularly in cross-city generalization. The paper also provides several ablation studies to demonstrate the effectiveness of its specific engineering designs, such as the backward planning and the competitive interaction.

**Compliance With Llm Reviewing Policy:**

Affirmed.

**Final Justification:**

I thank the authors for their thorough responses. I appreciate the added closed-loop NAVSIM result and the clearer explanation of how the competitive mechanism stabilizes RL under world-model bias. These clarifications improved my confidence in the framework. However, my concerns are only partially resolved. The new closed-loop evidence is certainly valuable and directionally encouraging, but it is still somewhat limited for fully validating the reactive driving capability of the method. Similarly, the additional discussion on hallucinations and value overestimation makes the mitigation mechanism much clearer, but in my view it does not yet fully resolve the underlying issue itself. I therefore maintain my original overall assessment.

**Key Questions For Authors:**

- **Closed-loop Evaluation**: Since Reinforcement Learning is fundamentally built upon the interactive feedback between policy and environment, open-loop metrics on nuScenes are insufficient to prove that the RL actor has truly learned **"collision avoidance"** rather than just overfitting to the fixed trajectories in the dataset. To demonstrate the model's actual performance in a reactive environment, could the authors provide **closed-loop** results (e.g., on Bench2Drive or NAVSIM)?

- OOD Hallucinations in the World Model: The introduction states that world models suffer from OOD hallucinations during RL exploration, leading to over-optimistic value estimates. However, beyond the "competitive mechanism", the paper does not directly explain how this issue is resolved. Could the authors clarify the specific approach used to address these hallucinations?

**Limitations:**

Although this paper does not include a dedicated section for limitations, several constraints and weaknesses are acknowledged throughout the text, particularly in the Appendix. The key limitations include:

- Bias from non-reactive simulation: The system employs non-reactive simulation where future states and rewards are generated by the world model. This dependency can introduce significant biases and hallucinations, which potentially limits the model's true generalization ability in real-world interactive scenarios.

- some bad case

**Strengths And Weaknesses:**

**Strengths:**

- Solid motivation and elegant integration: Marrying IL and RL for E2E autonomous driving makes a lot of practical sense. The ablation studies are convincing and show that the authors didn't just naively add them together; the integration is done quite elegantly.

- Strong empirical results: The collision rate metrics are very good. It is particularly impressive that the method actually works well in cross-city generalization scenarios, which is notoriously hard.

- Neat architectural trick: The "backward mask" (inverse causality) is a very nice touch. It's a simple, slightly counter-intuitive idea that clearly works well in practice.

**Weaknesses:**
See the Questions section.

---

> ### Author Rebuttal · Authors · 2026-03-31
>
> ### Q1: Close-loop Evaluation
> We fully agree that closed-loop evaluation is crucial to demonstrate reactive collision avoidance. As requested, we adapted our framework to NAVSIM. Please refer to **Exp3** below for detailed closed-loop results, which confirm our method's cross-framework applicability.
>
> ### Q2: OOD hallucination & Overestimate Values & Competitive Mechanism
> please see **Exp1** below to get diagnostic evidence on how our competitive mechanism mitigates value overestimation. To further clarify our approach:
> - **Value Overestimation:** While our rule-based reward $r(s,a)$ is unbiased, the Critic updates via TD learning: $V(s_t) \approx r + \gamma V(s_{t+1})$. When the World Model hallucinates $s_{t+1}$, it corrupts the value estimate. The RL actor exploits these over-optimistic values, risking policy collapse.
> - **Unbiased Evaluation & Recovery:** Our **Competitive Mechanism** evaluates both actors across full trajectories using *only* unbiased rewards, effectively bypassing the corrupted Critic. If the RL actor underperforms the stable IL baseline, we reset its parameters to the IL actor's. This confines RL exploration to a safe "trust zone"—allowing flexible deviation from strict expert demonstrations, while utilizing the IL policy as an anchor to instantly recover from catastrophic divergence.
> - **Summary:** Instead of trying to prevent world model hallucinations, we use the IL actor as a structural regularizer. It actively "resets" the RL policy back to the expert distribution whenever it is misled by OOD shortcuts.
>
> ### L1: Bias from Non-reactive Simulation
>
> We agree offline world models cannot perfectly replace high-fidelity simulators. We clarify our framework's role:
>
> 1. **World Model as a Reactive Proxy:** Unlike static "log-replay" in traditional IL, our **Latent World Model** learns a transition function $WM(s_{t+1}|s_t, a_t)$. This acts as a *reactive simulator* where future states are dynamic, enabling RL exploration.
> 2. **The Offline Data Constraint:** We acknowledge this reactivity is fundamentally constrained by nuScenes data, risking bias or hallucinations when the agent explores far from the expert distribution (OOD).
> 3. **Mitigation via Competition:** To guard against this, our **Competitive Mechanism** periodically anchors the RL policy to the stable IL policy. This prevents the model from "over-generalizing" into hallucinated, physically ungrounded latent regions.
>
> **In summary:** Our approach serves as a critical bridge. It provides a more reactive training signal than pure IL, while our competitive dual-policy design actively filters out the hallucinations inherent in offline transition modeling.
>
> ---
>
> ### Exp3 (for Q1): Closed-loop Results/Cross-framework Applicability/Generalizes to Other Offline Driving Datasets
>
> To prove our method's cross-framework applicability and closed-loop performance, we adapted CoIRL-AD to the SOTA Navsim v1 baseline (WoTE), utilizing PDM score components as our RL reward.
> - **Results:** Despite the short rebuttal window, integrating CoIRL-AD's core mechanisms onto the WoTE backbone improved the overall PDM Score (PDMS) from 87.9 to 88.2, while also increasing Drivable Area Compliance (DAC) and Time to Collision (TTC) metrics.
> - **Conclusion:** This confirms our approach is framework-agnostic. It successfully generalizes beyond nuScenes and the LAW architecture to alternative offline datasets and reactive environments.
>
> For full experimental setup, benchmark tables, and code, please refer to [exp3 **(click here)**](https://anonymous.4open.science/r/drive-with-two-minds/icml_rebuttal/exp3/README.md) in our anonymous repository.
>
>
> ### Exp1 (for Q2): Diagnostic Evidence for Competitive Mechanism
> To prove our competitive mechanism mitigates instability, we evaluated CoIRL-AD with and without it.
> - **Value Overestimation:** Without competition, the critic's value estimate for hallucinated future states $V(\hat{s}_{t+1})$ climbs uncontrollably. Our mechanism successfully plateaus this, preventing the exploitation of world model inaccuracies.
> - **Policy Divergence:** Unconstrained RL pursues these hallucinated rewards, causing severe policy shift (high L2 distance from the expert). The competitive mechanism constrains exploration to a "trust zone," pulling the RL actor back.
> - **Performance Impact:** This stabilization directly improves safety. With the mechanism active, average collision rates (Col@avg) drop from 0.29 to 0.18, and trajectory deviation (L2@avg) drops from 0.72 to 0.63.
>
>  For full training visualizations, loss curves, and raw logs, please see [exp1 **(click here)**](https://anonymous.4open.science/r/drive-with-two-minds/icml_rebuttal/exp1/README.md) in our anonymous repository.

---

> > ### Author Rebuttal · Reviewer_LjSX · 2026-04-03
> >
> > Thank you for the thoughtful and detailed rebuttal. I really appreciate that the authors took my concerns seriously and responded in a very concrete way, especially by adding a new closed-loop NAVSIM result and by providing a much clearer explanation of how the competitive mechanism helps stabilize RL under world-model bias.
> >
> > Overall, I found the rebuttal helpful and well prepared, and it did increase my confidence in the technical soundness of the proposed framework. That said, my concerns are only partially resolved rather than fully eliminated. The new closed-loop evidence is certainly valuable and directionally encouraging, but it is still somewhat limited for fully validating the reactive driving capability of the method. Similarly, the additional discussion on hallucinations and value overestimation makes the mitigation mechanism much clearer, but in my view it does not yet fully resolve the underlying issue itself. Since these concerns are tied to the core evaluation setting and the assumptions behind the learned world model, they are not easy to settle within a short rebuttal. I have carefully taken the rebuttal into account, but I am currently maintaining my original overall assessment.

---

> > > ### Author Response · Authors · 2026-04-05
> > >
> > > Dear Reviewer LjSX,
> > >
> > > Thank you for your thoughtful follow-up and for explicitly noting that our rebuttal increased your confidence in the technical soundness of our framework. We are glad that the new NAVSIM closed-loop results and the expanded explanation of our competitive mechanism were helpful and directionally encouraging.
> > >
> > > We completely agree with your assessment that our paper does not entirely "solve" the underlying, fundamental challenges of reactive closed-loop driving and world-model hallucinations. You have rightly highlighted two of the most critical open problems in the field today, and we view our work as a stepping stone rather than the final answer. We wanted to briefly share our perspective on how our work fits into these broader community challenges:
> > >
> > > **1. Validating Reactive Driving (The NAVSIM Proxy)**
> > > We agree that true, high-fidelity reactive simulation is the gold standard. However, as evaluating models in actual closed-loop simulators is often computationally prohibitive or framework-dependent, we specifically chose NAVSIM as our rebuttal proxy based on the authors' findings. As stated in the NAVSIM paper, the PDM Score (PDMS) is explicitly designed to bridge this gap: *"the PDMS is better correlated to the closed-loop score (CLS) than the open-loop score (OLS), and is always positively correlated."* While not a perfect substitute for full reactive deployment, we believe adapting our framework to achieve PDMS gains provides the strongest currently available proxy for closed-loop safety.
> > >
> > > **2. Evaluating Latent World Models**
> > > Your point regarding the underlying assumptions of the learned world model is highly insightful. Because our world model operates entirely in the latent space—optimized strictly for downstream control rather than pixel-level video generation—it is inherently difficult to "prove" the absence of hallucinations through direct visualization or pixel-level metrics. Following the precedent set by other latent architectures (e.g., V-JEPA), we must rely on the downstream success rate of the policy to validate the latent representations. Our competitive mechanism does not theoretically "cure" hallucinations; rather, it is a practical, architectural safeguard that prevents the RL agent from catastrophically exploiting them. Completely eliminating these hallucinations at the representation level remains an exciting open challenge for the community.
> > >
> > > Thank you again for pushing us to test on NAVSIM and for championing the importance of these rigorous evaluation standards. Your feedback will help us frame our limitations and future work sections much more effectively in the camera-ready version.

---

### Decision · Program_Chairs · 2026-04-30

**Decision:**

Accept (regular)

**Comment:**

This paper proposes CoIRL-AD, an offline end-to-end autonomous driving framework that combines imitation learning (IL) and reinforcement learning (RL) using a shared latent world model and two decoupled actors coupled via a competitive parameter exchange/reset mechanism. The goal is to improve robustness and long-tail performance without relying on an external simulator.

The main objections (primarily from Reviewer gMNL) concerned (i) unclear novelty and attribution across multiple design choices, (ii) potential selection bias in the long-tail evaluation, and (iii) limited closed-loop/reactive validation and limited direct evaluation of world-model failure modes. In the rebuttal/discussion, the authors added evidence: a symmetric long-tail subset construction to address selection bias, additional ablations clarifying that the gains arise from the synergy between the inverse-causal mask and the competitive mechanism (rather than either alone), and diagnostic evidence that the competitive mechanism mitigates value overestimation/policy drift under world-model hallucinations. They also reported a NAVSIM adaptation with a small but positive PDMS gain, partially addressing cross-framework applicability and closed-loop relevance.

These additions resolved most experimental concerns for three reviewers, who maintained Weak Accept, while the remaining reject largely argued that the amount of revision implied by the rebuttal warrants a full re-review cycle.

Overall, the remaining weaknesses appear to be presentation/completeness rather than a clear technical flaw: the manuscript would benefit from clearer framing of core contributions vs. design choices, stronger organization and writing polish, explicit limitations/impact discussion, and (if possible) stronger closed-loop validation given world-model hallucination risks.

Considering the rebuttal evidence and the reviewer majority, my recommendation is Weak Accept, conditional on incorporating the new analyses/ablations into the main paper and substantially improving clarity and reporting in the camera-ready version.